# On the Almost Sure Convergence of Stochastic Gradient Descent in Non-Convex Problems

**Panayotis Mertikopoulos**
Univ. Grenoble Alpes, CNRS, Inria, Grenoble INP, LIG &
Criteo AI Lab
panayotis.mertikopoulos@imag.fr

**Nadav Hallak**
Technion
ndvhllk@technion.ac.il

**Ali Kavis**
École Polytechnique Fédérale de Lausanne (EPFL)
ali.kavis@epfl.ch

**Volkan Cevher**
École Polytechnique Fédérale de Lausanne (EPFL)
volkan.cevher@epfl.ch

## Abstract

This paper analyzes the trajectories of stochastic gradient descent (SGD) to help understand the algorithm's convergence properties in non-convex problems. We first show that the sequence of iterates generated by SGD remains bounded and converges with probability 1 under a very broad range of step-size schedules. Subsequently, going beyond existing positive probability guarantees, we show that SGD avoids strict saddle points/manifolds with probability 1 for the entire spectrum of step-size policies considered. Finally, we prove that the algorithm's rate of convergence to local minimizers with a positive-definite Hessian is $\mathcal{O}(1/n^p)$ if the method is employed with a $\Theta(1/n^p)$ step-size. This provides an important guideline for tuning the algorithm's step-size as it suggests that a cool-down phase with a vanishing step-size could lead to faster convergence; we demonstrate this heuristic using ResNet architectures on CIFAR.

## 1 Introduction

Owing to its simplicity and empirical successes, stochastic gradient descent (SGD) has become the de facto method for training a wide range of models in machine learning. This paper examines the properties of SGD in non-convex problems with the aim of answering the following questions:

(Q1) Does SGD *always* converge to the problem's critical set?
(Q2) Does SGD *always* avoid spurious critical regions (non-isolated saddle points, ridges, etc.)?
(Q3) How fast does SGD converge to local minimizers as a function of the method's step-size?

We provide the following contributions to these questions:

**On (Q1):** Under mild conditions for the function to be optimized, and allowing for a wide range of step-size schedules of the form $\Theta(1/n^p)$ for $p \in (0, 1]$, the sequence of iterates $X_n$ generated by SGD converges with probability 1. In contrast to existing mean squared error guarantees of the form $\mathbb{E}[\|\nabla f(X_n)\|^2] \to 0$ (where $f$ is the problem's objective), our result is a stronger, trajectory convergence result: It is not a guarantee that holds on average, but a convergence certificate that applies with probability 1 to *any* instantiation of the algorithm.

**On (Q2):** With probablity 1, the trajectories of SGD avoid all strict saddle manifolds – i.e., sets of critical points $x^*$ with at least one negative Hessian eigenvalue ($\lambda_{\min}(\nabla^2 f(x^*)) < 0$). Such manifolds include ridge surfaces and other connected sets of non-isolated saddle points that are common in the loss landscapes of overparametrized neural networks [31]. In this way, our result complements and extends a series of almost sure saddle avoidance results for *deterministic* gradient descent [13, 14, 27, 28, 39], and with *high probability* [15] or *in expectation* [48] for SGD.

**On (Q3):** If SGD is run with a step-size schedule of the form $\gamma_n = \Theta(1/n^p)$ for some $p \in (0, 1]$, the algorithm enjoys a local convergence rate of the form $\mathbb{E}[f(X_n) - f(x^*)] = \mathcal{O}(1/n^p)$ relative to Hurwicz local minimizers (i.e., $\nabla^2 f(x^*) \succ 0$). We stress here that this is a "last iterate" convergence guarantee; neither ergodic, nor of a mean-squared gradient norm type. This is crucial for real-world applications because, in practice, SGD training is based on the last generated point.

Taken together, the above suggests that a vanishing step-size policy has significant theoretical benefits: almost sure convergence, avoidance of spurious critical points (again with probability 1), and fast stabilization to local minimizers. We explore these properties in a range of standard non-convex test functions and by training a ResNet architecture for a classification task over CIFAR.

The linchpin of our approach is the *ODE method* of stochastic approximation as pioneered by Benveniste et al. [5], Kushner and Yin [26], Ljung [32], and Benaïm [2]. As such, our analysis combines a wide range of techniques from the theory of dynamical systems along with a series of martingale limit theory tools originally developed by Pemantle [40] and Brandière and Duflo [9].

**Related work.** Ever since the seminal paper of Robbins and Monro [43], SGD has given rise to a vast corpus of literature that we cannot hope to do justice here. We discuss below only those works which – to the best of our knowledge – are the most relevant to the contributions outlined above.

The first result on the convergence of SGD trajectories is due to Ljung [32, 33], who proved the method's convergence under the boundedness assumption $\sup_n \|X_n\| < \infty$. Albeit intuitive, this assumption is fairly difficult to establish from first principles and the problem's primitives. Because of this, the boundedness of SGD trajectories has persisted in the stochastic approximation literature as a condition that needs to be enforced "by hand", see e.g., Benaïm [2], Borkar [8], Kushner and Yin [26], and references therein. Bonnabel [7] also proved a range of trajectory convergence results for SGD in Riemannian manifolds; however, the underlying assumption of this work was, again, that the algorithm's iterates somehow remain in a compact set. To rid ourselves of unverifiable requirements of this kind, we resort to a series of shadowing arguments that interpolate between continuous and discrete time. Our results also improve on a more recent result by Bertsekas and Tsitsiklis [6] who use a completely different analysis to dispense of boundedness via the use of more restrictive, Robbins–Monro step-size policy.

On the issue of saddle-point avoidance, Pemantle [40] and Brandière and Duflo [9] showed that SGD avoids *hyperbolic* saddle points ($\lambda_{\min}(\nabla^2 f(x^*)) < 0$, $\det \nabla^2 f(x^*) \neq 0$) with probability 1. More recently, and under different assumptions, Ge et al. [15] showed that SGD avoids *strict* saddle points ($\lambda_{\min}(\nabla^2 f(x^*)) < 0$) with high probability: specifically, Ge et al. [15] showed that, if run with a constant step-size $\gamma$, then, with probability at least $1 - \zeta$, SGD produces iterates that are close to a local minimizer (and hence away from saddle points) after $\Theta(\log(1/\zeta)/\gamma^2)$ iterations. Daneshmand et al. [12] further refined this result by obtaining positive probability results for *second-order* stationary points, whereas Vlaski and Sayed [48] guarantees escape from strict saddles in expectation. By comparison, our paper shows that strict saddles are avoided *with probability* 1, thus providing the missing link between these two complementary research threads; for completes, we review these results in detail in Section 4.3.

The papers mentioned above should be disjoined from an extensive literature on saddle-point avoidance results for *deterministic* gradient descent [13, 14, 24, 27, 28, 38, 39]. Given that these works focus exclusively on deterministic methods, they have no bearing on our results and any analogies between this literature and stochastic methods should be taken with a grain of salt. Indeed, in the case of deterministic gradient dynamics, there is a well-defined drift that consistently drives the method's iterates away from saddle points; in the stochastic case, this persistent push is no longer present, so the situation is considerably more complex.

Finally, regarding the rate of convergence of SGD in non-convex problems, Ghadimi and Lan [16, 17] established a series of bounds of the form $\mathbb{E}[\|\nabla f(X_R)\|^2] = \mathcal{O}(1/\sqrt{T})$, where $R$ is drawn randomly

from the running horizon $\{1, \ldots, T\}$ of the process. More recently, Lei et al. [30] provided a non-asymptotic rate analysis for $\alpha$-Holder smooth functions, without a bounded gradient assumption; specifically, Lei et al. [30] proved that, for some $T$, $\min_{n \leq T} \mathbb{E}[\|\nabla f(X_n)\|^2] = \mathcal{O}(T^{p-1})$ with stepsize $\gamma_n = \gamma/n^p$ and $p \in (1/(1+\alpha), 1)$. There is no overlap of our results or analysis with these works, and we are not aware of convergence guarantees similar to our own in the literature.

We should also stress that our analysis focuses exclusively on vanishing step-size policies, typically of the of the form $\gamma_n \propto 1/n^p$ for some $p \in (0, 1]$. The reason for this is simple: if run with a constant step-size, SGD trajectories *do not converge to the problem's critical set*, except in certain special cases (e.g., when the gradient noise vanishes over time). Indeed, under persistent noise, SGD with a constant step-size forms an irreducible ergodic Markov chain with an invariant measure that is supported over the entire search domain. In the best-case scenario, this means that SGD trajectories may be concentrated around the problem's critical set; however, they will never converge to it (in probability or almost surely). Because we are interested in questions of convergence, we will not treat constant step-size policies in the sequel.

## 2 Problem setup and assumptions

**2.1. Problem setup.** Throughout the sequel, we focus on the non-convex optimization problem

$$\text{minimize}_{x \in \mathbb{R}^d} \ f(x), \tag{Opt}$$

where $f \colon \mathbb{R}^d \to \mathbb{R}$ is a $d$-times differentiable function satisfying the following blanket assumptions.

**Assumption 1.** $f$ is *G-Lipschitz continuous and L-Lipschitz smooth*, i.e., for all $x, x' \in \mathbb{R}^d$, we have

$$|f(x') - f(x)| \leq G\|x' - x\|, \tag{1a}$$

$$\|\nabla f(x') - \nabla f(x)\| \leq L\|x' - x\|, \tag{1b}$$

**Assumption 2.** $f$ is *coercive*, i.e., $f(x) \to \infty$ as $\|x\| \to \infty$.

**Assumption 3.** $f$ is not *asymptotically flat*, i.e., $\liminf_{\|x\| \to \infty} \|\nabla f(x)\| > 0$.

Assumptions 1–3 are fairly standard in non-convex analysis and optimization. Taken individually, Assumption 1 is a basic regularity requirement for $f$; Assumption 2 guarantees the existence of solutions to (Opt) by ruling out vacuous cases like $f(x) = -x$; finally, Assumption 3 rules out functions like $f(x) = -e^{-x^2}$ that become near-critical at infinity. Taken together, Assumptions 1–3 further imply that the critical set

$$\mathcal{X}^* \equiv \mathrm{crit}(f) = \{x \in \mathbb{R}^d : \nabla f(x) = 0\} \tag{2}$$

of $f$ is nonempty, a fact that we use freely in the sequel.

**2.2. Assumptions on the oracle.** Typical examples of (Opt) in machine learning comprise neural networks with sigmoid activation functions, underdetermined inverse problems, empirical risk minimization models, etc. In such problems, obtaining accurate gradient input is impractical, so to solve (Opt), we often rely on *stochastic gradient* information, obtained for example by taking a mini-batch of training instances.

With this in mind, we will assume throughout that the optimizer can access $\nabla f$ via a *stochastic first-order oracle* (SFO). Formally, this is a black-box feedback mechanism which, when queried at an input point $x \in \mathbb{R}^d$, returns a random vector $\mathsf{V}(x; \omega)$ with $\omega$ drawn from some (complete) probability space $(\Omega, \mathcal{F}, \mathbb{P})$. In more detail, decomposing the oracle's output at $x$ as

$$\mathsf{V}(x; \omega) = \nabla f(x) + Z(x; \omega), \tag{SFO}$$

we make the following assumption.

**Assumption 4.** The *error term* $Z(x; \omega)$ of (SFO) has

(*a*) *Zero mean:* $\qquad\qquad \mathbb{E}[Z(x; \omega)] = 0$ $\qquad\qquad\qquad\qquad\qquad\qquad\qquad$ (3a)

(*b*) *Finite q-th moments:* $\quad \mathbb{E}[\|Z(x; \omega)\|^q] \leq \sigma^q$ for some $q \geq 2$ and $\sigma \geq 0$. $\qquad$ (3b)

Assumption 4 is standard in stochastic optimization and is usually stated with $q = 2$, i.e., as a "finite variance" condition, cf. Benaïm [2], Juditsky et al. [25], Nesterov [37], Polyak [42], and many others. Allowing values of $q$ greater than 2 provides more flexibility in the choice of step-size policies, so we keep (3b) as a blanket assumption throughout. We also formally allow the value $q = \infty$ in (3b), in which case we will say that the noise is *bounded in* $L^\infty$; put simply, this corresponds to the standard assumption that the noise in (SFO) is bounded almost surely.

**2.3. Stochastic gradient descent.**    With all this in hand, the stochastic gradient descent (SGD) algorithm can be written as

$$X_{n+1} = X_n - \gamma_n V_n. \tag{SGD}$$

In the above, $n = 1, 2, \ldots$ is the algorithm's iteration counter, $\gamma_n$ is the algorithm's step-size, and $V_n$ is a sequence of gradient signals of the form

$$V_n = \mathsf{V}(X_n; \omega_n) = \nabla f(X_n) + Z_n. \tag{4}$$

Each gradient signal $V_n$ is generated by querying the oracle at $X_n$ with some random seed $\omega_n$. For concision, we write $Z_n \equiv Z(X_n, \omega_n)$ for the gradient error at the $n$-th iteration and $\mathcal{F}_n = \sigma(X_1, \ldots, X_n)$ for the natural filtration of $X_n$; in this notation, $\omega_n$ and $V_n$ are *not* $\mathcal{F}_n$-measurable.

All our results for (SGD) are stated in the framework of the basic assumptions above. The price to pay for this degree of generality is that the analysis requires an intricate interplay between martingale limit theory and the theory of stochastic approximation; we review the relevant notions below.

# 3   Stochastic approximation

**Asymptotic pseudotrajectories.**    The departure point for our analysis is to rewrite the iterates of (SGD) as $(X_{n+1} - X_n)/\gamma_n = \nabla(f(X_n)) + Z_n$. In this way, (SGD) can be seen as a Robbins–Monro discretization of the continuous-time *gradient dynamics*

$$\dot{x}(t) = -\nabla f(x(t)). \tag{GD}$$

The main motivation for this comparison is that $f$ is a strict Lyapunov function for (GD), indicating that its solution orbits converge to the critical set $\mathcal{X}^*$ of $f$ (see the supplement for a formal statement and proof of this fact). As such, if the trajectories of (SGD) are "good enough" approximations of the solutions of (GD), one would expect (SGD) to enjoy similar convergence properties.

To make this idea precise, we first connect continuous and discrete time by letting $\tau_n = \sum_{k=1}^n \gamma_k$ denote the time that has "elapsed" for (SGD) up to iteration counter $n$ (inclusive); that is, a step-size in discrete time is translated to elapsed time in the continuous case, and vice-versa. We may then define the continuous-time interpolation of an iterate sequence $X_n$ of (SGD) as

$$X(t) = X_n + [(t - \tau_n)/(\tau_{n+1} - \tau_n)](X_{n+1} - X_n) \qquad \text{for all } t \in [\tau_n, \tau_{n+1}]. \tag{5}$$

To compare this trajectory to the solutions of (GD), we further need to define the "flow" of (GD) which describes how an ensemble of initial conditions evolves over time. Formally, we let $\Phi \colon \mathbb{R}_+ \times \mathbb{R}^d \to \mathbb{R}^d$ denote the map which sends an initial $x \in \mathbb{R}^d$ to the point $\Phi_t(x) \in \mathbb{R}^d$ by following for time $t \in \mathbb{R}_+$ the solution of (GD) starting at $x$. We then have the following notion of "asymptotic closeness" between a sequence generated by (SGD) and the flow of the dynamics (GD):

**Definition 1.**  We say that $X(t)$ is an *asymptotic pseudotrajectory* (APT) of (GD) if, for all $T > 0$:

$$\lim_{t \to \infty} \sup_{0 \le h \le T} \|X(t + h) - \Phi_h(X(t))\| = 0, \tag{6}$$

The notion of an APT is due to Benaïm and Hirsch [4] and essentially posits that $X(t)$ tracks the flow of (GD) with arbitrary accuracy over windows of arbitrary length as $t \to \infty$. When this is the case, we will slightly abuse terminology and say that the sequence $X_n$ itself comprises an APT of (GD).

With all this in hand, the formal link between (GD) and (SGD) is as follows:

**Proposition 1.**  *Suppose that Assumptions 1 and 4 hold and (SGD) is employed with a step-size sequence such that $\sum_{n=1}^\infty \gamma_n = \infty$ and $\sum_{n=1}^\infty \gamma_n^{1+q/2} < \infty$ with $q \ge 2$ as in Assumption 4. Then, with probability 1, $X_n$ is an APT of (GD).*

**Corollary 1.**  *Suppose that (SGD) is run with $\gamma_n = \Theta(1/n^p)$ for some $p \in (2/(q+2), 1]$ and assumptions as in Proposition 1. Then, with probability 1, $X_n$ is an APT of (GD).*

This comparison result plays a key role in the sequel because it delineates the range of step-size policies under which the discrete-time system (SGD) is well-approximated by the continuous-time dynamics (GD). We discuss this issue in detail in the next section.

# 4  Convergence analysis

Heuristically, the goal of approximating (SGD) via (GD) is to reduce the difficulty of the direct analysis of the former by leveraging the strong convergence properties of the latter. of the latter, which is relatively straightforward to analyze, to the former, which is much more difficult. However, the notion of an APT does not suffice in this regard: in the supplement, we provide an example where a discrete-time APT has a completely different behavior relative to the underlying flow. As such, a considerable part of our analysis below focuses on tightening the guarantees provided by the APT approximation scheme.

**4.1. Boundedness and stability of the approximation.**   The basic point of failure in the stochastic approximation approach is that APTs may escape to infinity, rendering the whole scheme useless, cf. [2, 8] and references therein. It is for this reason that a large part of the literature on SGD explicitly assumes that the trajectories of the process are bounded (precompact), i.e.,

$$\sup_{t \geq 0} \|X(t)\| < \infty \quad (a.s.). \tag{7}$$

However, this is a prohibitively strong assumption for (Opt): unless certified ahead of time, any theoretical result relying on this assumption would be of limited practical value.

Our first result below provides exactly this certification by establishing that (7) is solely an implication of our underlying Assumptions 1–4. It is a non-trivial outcome which provides the key to unlocking the potential of stochastic approximation techniques in the sequel.

**Theorem 1.** *Suppose that Assumptions 1–4 hold and (SGD) is run with a variable step-size sequence of the form $\gamma_n \propto 1/n^p$ for some $p \in (2/(q+2), 1]$. Then, with probability 1, every APT $X(t)$ of (GD) that is induced by (SGD) has $\sup_{t \geq 0} \|X(t)\| < \infty$.*

Because of the generality of our assumptions, the proof of Theorem 1 involves a delicate combination of non-standard techniques; for completeness, we provide a short sketch below and refer the reader to the supplement for the details.

*Sketch of proof of Theorem 1.*  The main reasoning evolves along the following lines:

Step 1.  We first show that, under the stated assumptions, there exists a (possibly random) subsequence $X_{n_k}$ of $X_n$ that converges to $\mathcal{X}^*$; formally, $\liminf_{n \to \infty} \operatorname{dist}(X_n, \mathcal{X}^*) = 0$ (a.s.). As a result, $X(t)$ eventually reaches a sublevel set $L_\varepsilon$ whose elements are arbitrarily close to $\mathcal{X}^*$, i.e., there exists some $t_\varepsilon > 0$ such that $X(t_\varepsilon) \in L_\varepsilon$.

Step 2.  By a technical argument relying on the regularity assumptions for $f$ (cf. Assumption 1), it can be shown that there exists some *uniform* time window $\tau$ such that $X(t)$ remains within uniformly bounded distance to $L_\varepsilon$ for all $t \in [t_\varepsilon, t_\varepsilon + \tau]$. Thus, once $X(t)$ gets close to $L_\varepsilon$, it will not escape too far within a fixed length of time.

Step 3.  An additional technical argument reveals that, under the stated assumptions for $f$, the trajectories of (GD) either descend the objective by a uniform amount, or they have reached a neighborhood of the critical set $\mathcal{X}^*$ where further descent is impossible (or irrelevant).

Step 4.  By combining the two previous steps, we conclude that $X(t_\varepsilon + \tau) \in L_\varepsilon$ at the end of said window. This argument may then be iterated ad infinitum to show inductively that $X(t) \in L_\varepsilon$ for all intervals of the form $[t_\varepsilon + k\tau, t_\varepsilon + (k+1)\tau]$.

Since $L_\varepsilon$ is bounded (by Assumption 2), we conclude that $X(t)$ remains in a compact set for all $t \geq 0$, i.e., $X(t)$ is precompact. The conclusion of Theorem 1 then follows by Corollary 1.   ∎

**4.2. Almost sure convergence.**   By virtue of Theorem 1, we are now in a position to state our almost sure convergence result:

**Theorem 2.** *Suppose that Assumptions 1–4 hold and (SGD) is run with a variable step-size sequence of the form $\gamma_n = \Theta(1/n^p)$ for some $p \in (2/(q+2), 1]$. Then, with probability 1, $X_n$ converges to a (possibly random) connected component $\mathcal{X}_\infty^*$ of $\mathcal{X}^*$ over which $f$ is constant.*

**Corollary 2.** *With assumptions as in Theorem 2, we have the following:*

    *1. $f(X_n)$ converges (a.s.) to some critical value $f_\infty$.*

    *2. Any limit point of $X_n$ is (a.s.) a critical point of $f$.*

Theorem 2 extends a range of existing treatments of (SGD) under explicit boundedness assumptions of the form (7), cf. [2, 8, 32] and references therein. It also improves on a similar result by Bertsekas and Tsitsiklis [6] who use a completely different analysis to dispense of boundedness requirements via the use of more restrictive step-size policies. Specifically, Bertsekas and Tsitsiklis [6] require the Robbins–Monro summability conditions $\sum_n \gamma_n = \infty$ and $\sum_n \gamma_n^2 < \infty$ under a bounded variance assumption. In this regard, our analysis extends to more general step-size policies, while that of Bertsekas and Tsitsiklis [6] cannot because of its reliance on the Robbins-Siegmund theorem for almost-supermartingales [44]. Among other benefits, this added degree of flexibility is a key advantage of the APT approach.

The heavy lifting in the proof of Theorem 2 is provided by Theorem 1. Thanks to this boundedness certificate, the total chain of implications is relatively short, so we provide it in full below.

*Proof of Theorem 2.* Under the stated assumptions, $f$ is a strict Lyapunov function for (GD) in the sense of Benaïm [2, Chap. 6.2]. Specifically, this means that $f(\Phi_t(x))$ is strictly decreasing in $t$ unless $x$ is a stationary point of (GD). Furthermore, by Sard's theorem [36, Chap. 2], the set $f(\mathcal{X}^*)$ of critical values of $f$ has Lebesgue measure zero – and hence, empty topological interior. Therefore, applying Theorem 5.7 and Proposition 6.4 of Benaïm [2] in tandem, we conclude that any precompact asymptotic pseudotrajectory of (GD) converges to a connected component $\mathcal{X}_\infty^*$ of $\mathcal{X}^*$ over which $f$ is constant. Since Theorem 1 guarantees that the APTs of (GD) induced by (SGD) are bounded with probability 1, our claim follows. ∎

**4.3. Avoidance analysis.** Theorem 2 represents a strong convergence guarantee but, at the same time, it does not characterize the component of $\mathcal{X}^*$ to which $X_n$ converges. The rest of this section is devoted to showing that $X_n$ does not converge to a component of $\mathcal{X}^*$ that only consists of saddle points (a *saddle-point manifold*). Specifically, we will make precise the following informal statement:

    (SGD) *avoids strict saddles – and sets thereof – with probability* 1.

To set the stage for the analysis to come, we begin by reviewing some classical and recent results on the avoidance of saddle points. We then present our general results towards the end of the section.

To begin, a crucial role will be played in the sequel by the *Hessian matrix* of $f$, viz.

$$H(x) \equiv \nabla^2 f(x) \equiv (\partial_i \partial_j f(x))_{i,j=1,\dots,d}. \tag{8}$$

Since $H(x)$ is symmetric, all of its eigenvalues are real. If $x^*$ is a critical point of $f$ and $\lambda_{\min}(H(x^*)) < 0$, we say that $x^*$ is a *strict saddle point* [27, 28].

By standard results in center manifold theory [47], the space around strict saddle points admits a decomposition into a *stable*, *center* and *unstable* manifold (each of the former two possibly of dimension zero; the latter of dimension at least 1 given that $\lambda_{\min}(H(x^*)) < 0$). Heuristically, under the continuous-time dynamics (GD), directions along the stable manifold of $x^*$ are attracted to $x^*$ at a linear rate, while those along the unstable manifold are repelled (again at a linear rate); the dynamics along the center manifold could be considerably more complicated, but, in the presence of unstable directions, they only emerge from a measure zero of initial conditions. As a result, if $x^*$ is a strict saddle point of $f$, it stands to reason that (SGD) should "probably" avoid it as well.

In the case of *deterministic* gradient descent with step-size $\gamma < 1/L$, this intuition was made precise by Lee et al. [27, 28] who proved that all but a measure zero of initializations of gradient descent avoid strict saddles. As we discussed in the introduction, this result was then extended to various deterministic settings, with different assumptions for the gradient oracle, the method's step-size, or the structure of the saddle-point manifold, see e.g., [13, 14, 24, 28, 38, 39] and the references therein.

In the stochastic regime, the situation is considerably more involved. Pemantle [40] and Brandière and Duflo [9] were the first to establish the avoidance of hyperbolic unstable equilibria in general stochastic approximation schemes. However, a key requirement in the analysis of these works is that of *hyperbolicity*, which in our setting amounts to asking that $H(x^*)$ is *invertible*. In particular,

this means the saddle point in question cannot be isolated, nor can it have a center manifold: both hypotheses are too stringent for applications of SGD to contemporary machine learning models, such as deep net training, so their results do not apply in many cases of practical interest.

More relevant for our purposes is the recent result of Ge et al. [15], who provided the following guarantee. Suppose that $f$ is $(\alpha, \beta, \varepsilon, \delta)$-*strict saddle*, i.e., for all $x \in \mathbb{R}^d$, one of the following holds: *(i)* $\|\nabla f(x)\| \geq \varepsilon$; *(ii)* $\lambda_{\min}(H(x)) \leq -\beta$; or *(iii)* $x$ is $\delta$-close to a local minimum $x_c$ around which $f$ is $\alpha$-fastly convex. Suppose further that $f$ is bounded, $L$-Lipschitz smooth, and $H(x)$ is $\rho$-Lipschitz continuous; finally, assume that the noise in the gradient oracle (SFO) is finite (a.s.) and contains a component uniformly sampled from the unit sphere. Then, given a confidence level $\zeta > 0$, and assuming that (SGD) is run with *constant* step-size $\gamma = \mathcal{O}(1/\log(1/\zeta))$, the algorithm produces after a given number of iterations a point which is $\mathcal{O}(\sqrt{\gamma \log(1/(\gamma\zeta))})$-close to $x_c$, and hence away from any strict saddle of $f$, with probability at least $1 - \zeta$.

In a more recent paper, Vlaski and Sayed [48] examined the convergence of (SGD) to second-order stationary points. More precisely, they showed that (SGD) guarantees *expected* descent for strict saddle points in a finite number of iterations, and with high probability, (SGD) iterates reach a set of *approximate* second-order stationary points in finite time.

The theory of Pemantle [40] and the result of Ge et al. [15] paint a complementary picture to the above: Pemantle [40] shows that saddle points are avoided with probability 1, provided they are hyperbolic (i.e., $\det \nabla^2 f(x^*) \neq 0$); on the other hand, Ge et al. [15] require much less structure on the saddle point, but they only provide a result with high probability (and $\zeta$ cannot be taken to zero because the range of allowable step-sizes would also vanish).[1] Our objective in the sequel is to provide a result that combines the "best of both worlds", i.e., almost sure avoidance of strict saddle points (and sets thereof) with probability 1.

To that end, we make the following assumption for the noise:

**Assumption 5.** The error term $Z \equiv Z(x; \omega)$ of (SFO) is *uniformly exciting*, i.e., there exists some $c > 0$ such that

$$\mathbb{E}[\langle Z(x; \omega), u \rangle^+] \geq c \tag{9}$$

for all $x \in \mathbb{R}^d$ and all unit vectors $u \in \mathbb{S}^{d-1}$.

This assumption simply means that the average projection of the noise along every ray in $\mathbb{R}^d$ is uniformly positive; in other words, $Z$ "excites" all directions uniformly – though not necessarily *isotropically*. As such, Assumption 5 is automatically satisfied by noisy gradient dynamics (e.g., as in Ge et al. [15]), generic finite sum objectives with at least $d$ summands, etc.

With all this in hand, we say that $\mathcal{S}$ is a *strict saddle manifold* of $f$ if it is a smooth connected component of $\mathcal{X}^*$ such that:

1. Every $x^* \in \mathcal{S}$ is a strict saddle point of $f$ (i.e., $\lambda_{\min}(H(x^*)) < 0$).
2. There exist $c_-, c_+ > 0$ such that, for all $x^* \in \mathcal{S}$, all negative eigenvalues of $H(x^*)$ are bounded from above by $-c_- < 0$, and any positive eigenvalues (*if* they exist) are bounded from below by $c_+$.

Somewhat informally, the definition of a strict saddle manifold implies that the eigenspaces of $H(x^*)$ corresponding to zero, positive, and negative eigenvalues decompose smoothly along $\mathcal{S}$ and $\mathcal{S}$ can be seen as an "integral manifold" of the nullspace of the Hessian of $f$.

With all this in hand, we are finally in a position to state our main avoidance result.

**Theorem 3.** *Suppose that* (SGD) *is run with a variable step-size sequence of the form* $\gamma_n \propto 1/n^p$ *for some* $p \in (0, 1]$. *If Assumptions 1–5 hold (with* $q = \infty$ *for Assumption 4), and* $\mathcal{S}$ *is a strict saddle manifold of* $f$, *we have* $\mathbb{P}(X_n \to \mathcal{S}$ *as* $n \to \infty) = 0$.

Theorem 3 is the formal version of the avoidance principle that we stated in the beginning of this section. Importantly, it makes *no* assumptions regarding the initialization of (SGD) and holds for *any* initial condition.

The proof of Theorem 3 relies on two basic components. The first is a probabilistic estimate, originally due to Pemantle [40], that shows that a certain class of stochastic processes avoids zero

with probability 1. The second is a differential-geometric argument in the spirit of Benaïm and Hirsch [3] and Benaïm [2], which uses center manifold theory to isolate the center/stable and unstable manifolds of $\mathcal{S}$. Combining these two components, it is possible to show that even ambulatory random walks along the stable manifold of $\mathcal{S}$ will eventually be expelled from a neighborhood of $\mathcal{S}$.

However, we should stress here that this point is quite subtle: the work of Pemantle [40] only applies to isolated, linearly unstable saddle points, and it does not cover saddle points with a non-trivial center manifold. In the deterministic case, strict saddles can indeed be excluded thanks to the existence of local diffeomorphism results based on the center manifold theorem. By contrast, in the stochastic case, the presence of a non-trivial center manifold requires fundamentally different techniques from differential geometry (cf. Appendices C.2 and C.3). The reason for this is that there is no longer a persistent drift away from the center stable manifold (in technical terms, there is no "shadowing"). This major difficulty is not present in Pemantle's analysis (which, again, cannot deal with non-trivial center manifolds); the only relation with [40] is two technical lemmas on random numerical sequences (Lemmas C.1 and C.2). We make all this precise in the paper's supplement.

**4.4. Rate of convergence.**    We conclude our analysis of (SGD) by establishing the algorithm's rate of convergence. Since $f$ is non-convex, any convergence rate analysis of this type must be a fortiori local; in view of this, we will examine the algorithm's convergence to local minimizers $x^* \in \mathcal{X}^*$ that are *regular* in the sense of Hurwicz, i.e., $H(x^*) \succ 0$. As we show in the supplement, a key property of regular minimizers is the following local strong convexity/smoothness lemma:

**Lemma 1.** *Let $x^*$ be a regular minimizer of $f$. Then, there exists a convex compact neighborhood $\mathcal{K}$ of $x^*$ and constants $\alpha, \beta > 0$ (possibly depending on $\mathcal{K}$) such that*

$$\alpha \|x - x^*\|^2 \le f(x) - f(x^*) \le \langle \nabla f(x), x - x^* \rangle \le \beta \|x - x^*\|^2 \quad \text{for all } x \in \mathcal{K}. \tag{10}$$

We are now in a position to state our main result for the rate of convergence of (SGD):

**Theorem 4.** *Fix some tolerance level $\delta > 0$, let $x^*$ be a regular minimizer of $f$, and suppose that Assumption 4 holds. Assume further that (SGD) is run with a step-size schedule of the form $\gamma_n = \gamma/(n + m)^p$ for some $p \in (2/(q + 2), 1]$ and large enough $m, \gamma > 0$. Then:*

1. *There exist neighborhoods $\mathcal{U}$ and $\mathcal{U}_1$ of $x^*$ such that, if $X_1 \in \mathcal{U}_1$, the event*

$$E_\infty = \{X_n \in \mathcal{U} \text{ for all } n = 1, 2, \dots\} \tag{11}$$

   *occurs with probability at least $1 - \delta$, i.e., $\mathbb{P}(E_\infty \mid X_1 \in \mathcal{U}_1) \ge 1 - \delta$.*

2. *Conditioned on $E$, we have*

$$\mathbb{E}[f(X_n) - f(x^*) \mid E_\infty] \le \frac{2}{\beta} \frac{\gamma}{1 - \delta} \frac{G^2 + \sigma^2}{2\alpha} \frac{1}{n^p} + o\left(\frac{1}{n^p}\right) \qquad \text{if } p < 1, \tag{12a}$$

   *and*

$$\mathbb{E}[f(X_n) - f(x^*) \mid E_\infty] \le \frac{2}{\beta} \frac{2\gamma^2}{1 - \delta} \frac{G^2 + \sigma^2}{2\alpha\gamma - 1} \frac{1}{n} + o\left(\frac{1}{n}\right) \qquad \text{if } p = 1, \tag{12b}$$

   *provided that $2\alpha\gamma > 1$ when $p = 1$.*

*Remark* 1. Note that Theorem 4 does not presuppose Assumptions 1–3; since the rate analysis is local, the differentiability of $f$ suffices. We should also stress here that the guarantees (12) differ substantially from other results in the literature [12, 16, 17, 24], in both scope and type, as they do not concern the ergodic average $\bar{X}_n = n^{-1} \sum_{k=1}^{n} X_k$ or the "best iterate" $X_n^{\text{best}} = \arg\min_{k=1,\dots,n} \|\nabla f(X_k)\|$ of (SGD). The former has relatively slow convergence in non-convex settings – typically $\mathcal{O}(1/\sqrt{n})$ if at all – while the latter cannot be calculated without access to perfect gradient information for the entire run of the process (in which case the stochastic element of SGD becomes meaningless).

*Remark* 2. It is also instructive to compare the rate guarantee (12b) to that of Ge et al. [15] and similar results. Theorem 4 explicitly requires a minimizer around which $f$ is locally strongly convex, and provides an $\mathcal{O}(1/n)$ value convergence rate if (SGD) is run with a $1/n$ step-size schedule. By contrast, Ge et al. [15] assumes (SGD) is run with a constant step-size $\gamma$, and requires $\mathcal{O}(1/\gamma^2)$ iterations to reach within $\mathcal{O}(\sqrt{\gamma})$ of a local minimizer. By the smoothness of $f$, this translates to an $\mathcal{O}(1/\sqrt{n})$ value convergence rate; morever, the definition of a "strict saddle function" in Ge et al. [15] implicitly assumes that $f$ is strongly convex in a neighborhood of said local minimizer. In this regard, the rate guarantee of Theorem 4 improves on that of Ge et al. [15] by an order of $\sqrt{n}$.

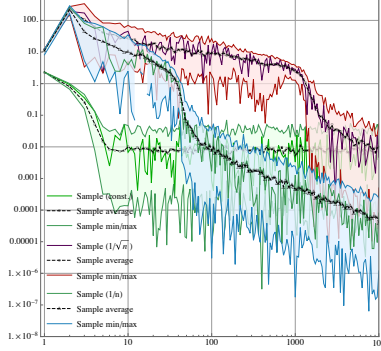

(a) Speed of convergence in the Shekel benchmark.

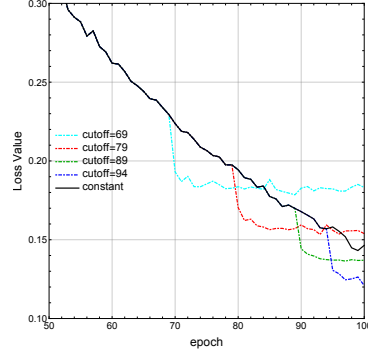

(b) Training ResNet with a cooldown heuristic.

The proof of Theorem 4 relies on showing that the distance to a regular minimizer is an almost supermartingale, *conditioned* on the fact that $X_n$ remains in a neighborhood of $x^*$. This conditioning poses a major challenge: a single bad realization of the noise can throw the algoritm out of the region of attraction of $x^*$, in which case there is no control over how $X_n$ may evolve. For this reason, we first need to show that *a*) $x^*$ is *stochastically stable*, i.e., with high probability, any initialization that is close enough to $x^*$ remains close enough [34, 35, 50]; and *b*) conditioned on this event, the distance $D_n = (1/2)\|X_n - x^*\|^2$ to a regular local minimizers behaves as an "almost" supermartingale [11, 44]. A major complication that arises here is that this changes the statistics of the noise, so the martingale property ceases to hold. Overcoming this difficulty requires a local probabilistic argument in the spirit of [20–22, 49, 50]; we provide the details in the supplement.

## 5 Numerical experiments

As an illustration of our theoretical analysis, we plot in Fig. 1a the convergence rate of (SGD) in the standard Shekel risk benchmark function $f(x) = \sum_{i=1}^{N} \left[ \sum_{j=1}^{d} (x_j - a_{ij})^2 + c_i \right]^{-1}$ where $A = (a_{ij})$ is a skew data matrix and $c = (c_1, \dots, c_N)$ is a bias vector of dimension $d = 500$ [23]. For our experiments, we ran $N = 10^3$ instances of (SGD) with a constant, $1/\sqrt{n}$, and $1/n$ step-size schedule, and we plotted the value difference $f(X_n) - f_\infty$ of the sample average (marked black lines) and the min-max spread of the samples for a 95% confidence level region (shaded green, red and blue respeciely for the constant, $1/\sqrt{n}$ and $1/n$ policies respectively). The constant step-size schedule initially performs better, but quickly saturates and is overcome by the $1/n$ schedule; overall, the $1/n$ policy converges faster than the other two by 2 to 4 orders of magnitude.

These tests suggest that a vanishing step-size policy may have significant advantages when used for training machine learning models. Of course, by itself, a rapidly vanishing step-size could cause the algorithm to traverse the loss landscape at a very slow pace and/or get trapped at inferior local minima. However, it also provides a sound theoretical justification for a "best of both worlds" heuristic: given a budget of gradient iterations, run SGD with a constant step-size for a fraction of this budget, and then implement a "cooldown" phase with a vanishing step-size. We demonstrate the benefits of this "cooldown" heuristic in a standard ResNet18 architecture for a classification task over CIFAR10. In particular, in Fig. 1b, we ran (SGD) with a constant step-size for 100 epochs, with checkpoints at different cutoffs; then, at each checkpoint, we launched the "cooldown" period with step-size $1/n$. Figure 1b demonstrates the improvement due to the cooldown period over the training loss: specifically, it shows that it is always beneficial to run the last epochs with a vanishing step-size.

## 6 Concluding remarks

Our aim in this paper was to present a novel trajectory-based analysis of (SGD) showing that, under minimal assumptions, *(i)* all of its limit points are stationary; *(ii)* it avoids strict saddle manifolds with probability 1; and *(iii)* it converges at a fast $\mathcal{O}(1/n)$ rate to regular minimizers. This opens the door to many interesting directions – from constrained/composite problems to adaptive gradient methods. We defer these to the future.

## Acknowledgments

This research was supported by the COST Action CA16228 "European Network for Game Theory" (GAMENET) and was partially conducted when the first author was visiting EPFL.

P. Mertikopoulos is grateful for financial support by the French National Research Agency (ANR) in the framework of the starting grant ORACLESS (ANR–16–CE33–0004–01), the "Investissements d'avenir" program (ANR-15-IDEX-02), the LabEx PERSYVAL (ANR-11-LABX-0025-01), and MIAI@Grenoble Alpes (ANR-19-P3IA-0003).

The work of N. Hallak was conducted at EPFL, and was supported by the European Research Council (ERC) under the European Union's Horizon 2020 research and innovation programme (grant agreement no. 725594 - time-data).

A. Kavis acknowledges financial support by the European Research Council (ERC) under the European Union's Horizon 2020 research and innovation programme (grant agreement n° 725594 - time-data).

V. Cevher gratefully acknowledges the support of the Swiss National Science Foundation (SNSF) under grant no. 200021–178865/1, the European Research Council (ERC) under the Horizon 2020 research and innovation programme (grant agreement No 725594 - time-data), and 2019 Google Faculty Research Award.

## Broader Impact

This work does not present any foreseeable societal consequence.

## Footnotes

[1]Pemantle [40] employs a vanishing step-size, which is more relevant for us: (SGD) with persistent noise and a constant step-size is an irreducible ergodic Markov chain whose trajectories do not converge *anywhere* [2].

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
