[Supplementary Material]

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

[2]That this is so is a consequence of the fact that the trajectories of (B.1) intersect the line $x_2 = 1$ at a vanishing angle as $t \to \infty$. More precisely, if we show the statement in question for $t_0$, it will also hold for all $\tau \geq t_0$ by virtue of the monotonicity of the exponential function.

[3]That such a set exists follows from the fact that $H(x^*)$ is symmetric.

[4]Recall here that a function $\phi$ has a right derivative when the limit $\nabla^+\phi(x)[v] \equiv \lim_{t\to 0^+}[\phi(x+tv) - \phi(x)]/t$ exists for all $v \in \mathbb{R}^d$.

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

# A  Convergence in continuous time

For completeness, we begin with a proof of the convergence of (GD) under our blanket assumptions:

**Proposition A.1** (Gradient flow convergence). *Under Assumptions 1 and 2, every solution $x(t)$ of* (GD) *converges to $\mathcal{X}^*$.*

*Proof of Proposition A.1.* To begin, existence and uniqueness of (global) solutions to (GD) follows readily from the Picard–Lindelöf theorem [47] and Assumption 1. With this point settled, and given that $f$ is coercive (cf. Assumption 2), the fact that $f$ is non-increasing along the orbits of (GD) shows that $x(t)$ converges to some compact invariant set $\mathcal{K} \subseteq \mathbb{R}^d$.

Suppose now that there exists a sequence of times $t_n$, $n = 1, 2, \ldots$, such that $x(t_n)$ converges to some *non-critical* point $\hat{x} \notin \mathrm{crit}(f) \equiv \mathcal{X}^*$. Letting $c = \|\nabla f(\hat{x})\|^2 > 0$, there exists a neighborhood $\mathcal{U}$ of $\hat{x}$ such that $\|\nabla f(x)\|^2 \geq c/2$ for all $x \in \mathcal{U}$ (again, by Assumption 1). Hence, by passing to a subsequence if necessary, we can assume without loss of generality that $x(t_n) > c/2$ for all $n$. Furthermore, by Assumption 1 (which implies that $\|\nabla f(x)\| \leq G$ for all $x \in \mathbb{R}^d$) and the definition of (GD), we have

$$\|x(t_n + \tau) - x(t_n)\| \leq \int_{t_n}^{t_n + \tau} \|\nabla f(x(s))\| \, ds \leq G\tau \tag{A.1}$$

for all $\tau > 0$. Therefore, by picking $\tau$ sufficiently small, we can assume that $x(t) \in \mathcal{U}$ for all $t \in [t_n, t_n + \tau]$ and all $n$ (recall here that $x(t_n) \in \mathcal{U}$ for all $n$). Then, by the definition of $\mathcal{U}$, we readily get

$$f(x(t_n + \tau)) - f(x(t_n)) = \int_{t_n}^{t_n + \tau} \frac{d}{dt} f(x(s)) \, ds = -\int_{t_n}^{t_n + \tau} \|\nabla f(x(s))\|^2 \, ds \leq -\frac{c\tau}{2}, \tag{A.2}$$

and hence:

$$f(x(t_n + \tau)) - f(x(0)) = -\int_0^{t_n + \tau} \|\nabla f(x(s))\|^2 \, ds \leq -\sum_{k=1}^{n} \int_{t_k}^{t_k + \tau} \|\nabla f(x(s))\|^2 \, ds$$

$$\leq -\sum_{k=1}^{n} \frac{c\tau}{2} = -\frac{nc\tau}{2} \tag{A.3}$$

i.e., $\lim_{n \to \infty} f(x(t_n + \tau)) = -\infty$, a contradiction. Since $x(t)$ converges to a compact invariant set $\mathcal{K}$, we conclude that $x(t)$ in fact converges to the critical set $\mathcal{X}^*$ of $f$. ∎

# B  Stability and boundedness of APTs

**B.1. Discrepancies between flows and APTs.**  Our first goal in this appendix is to provide a concrete example where asymptotic pseudotrajectories and the underlying continuous-time flow exhibit qualitatively different behaviors in the long run. To that end, consider the autonomous ODE

$$\dot{x}_1(t) = 1 \qquad \dot{x}_2(t) = -\frac{x_2(t)}{1 + x_1(t)} \tag{B.1}$$

which is a pseudo-gradient flow of the function $f(x_1, x_2) = x_2^2/2 - x_1$. The general solution of this system with initial condition $(0, b)$ at time $t = 0$ is

$$x(t) = \left( t, \frac{b}{1 + t} \right). \tag{B.2}$$

As a result, we have $x_2(t) \to 0$ as $t \to \infty$ from any initial condition (for a graphical illustration, see Fig. 2).

On the other hand, as we show below, the "constant height" curve $X(t) = (t, 1)$ is an asymptotic pseudotrajectory of (B.1). To show this, fix some accuracy threshold $\varepsilon > 0$ and a horizon $T > 0$. Then, with respect to Definition 1, it suffices to show that, for some sufficiently large $t_0 > 0$ and all $h \in [0, T]$, we have

$$|1 - x_2(t_0 + h)| \leq \varepsilon \tag{B.3}$$

**Figure 2:** Flowlines vs. asymptotic pseudotrajectories: the dashed black line is an APT of the depicted gradient flow but it stays at a constant height ($x_2 = 1$), even though all flow lines converge to the $x_1$-axis ($x_2 = 0$).

for the solution trajectory $x(t) = (x_1(t), x_2(t))$ that passes through the point $(t_0, 1)$ at time $t_0$.[2]

Substituting in the general solution of (B.1) and backsolving, we readily obtain that this trajectory has

$$x_2(t) = \frac{1 + t_0}{1 + t}. \tag{B.4}$$

In turn, this implies that the maximal difference between 1 and $x_2(t)$ over a window of size $T$ starting at $t_0$ is

$$\max_{0 \leq h \leq T} |1 - x_2(t_0 + h)| = \frac{1 + t_0 + T}{1 + t_0} - 1 = \frac{T}{1 + t_0} \leq \varepsilon \tag{B.5}$$

if $t_0$ is chosen sufficiently large – specifically, if $t_0 \geq T/\varepsilon - 1$.

Since $\varepsilon$ is arbitrary, the above shows that the APT condition (6) holds for all $T > 0$, i.e., $X(t)$ is an APT of (B.1). On the other hand, we have $\lim_{t\to\infty} X_2(t) = 1$, which is different than the limit of *any* solution of (B.1).

**B.2. Boundedness of APTs.**    Our aim in the rest of this appendix will be to prove Theorem 1, which, for convenience, we restate below:

**Theorem 1.** *Suppose that Assumptions 1–4 hold and (SGD) is run with a variable step-size sequence of the form $\gamma_n \propto 1/n^p$ for some $p \in (2/(q+2), 1]$. Then, with probability 1, every APT $X(t)$ of (GD) that is induced by (SGD) has $\sup_{t \geq 0} \|X(t)\| < \infty$.*

To begin, we recall the basic APT property of (SGD):

**Proposition 1.** *Suppose that Assumptions 1 and 4 hold and (SGD) is employed with a step-size sequence such that $\sum_{n=1}^{\infty} \gamma_n = \infty$ and $\sum_{n=1}^{\infty} \gamma_n^{1+q/2} < \infty$ with $q \geq 2$ as in Assumption 4. Then, with probability 1, $X_n$ is an APT of (GD).*

The proof of Proposition 1 follows by a tandem application of Propositions 4.1 and 4.2 of Benaïm [2], so we omit it; instead, we focus directly on the proof of Theorem 1. To that end, as we explained in the main body of the paper, the first part of our proof consists of showing that (SGD) admits a subsequence converging to $\mathcal{X}^*$, i.e., that $\liminf_{n\to\infty} \operatorname{dist}(X_n, \mathcal{X}^*) = 0$:

**Lemma B.1.** *With assumptions as in Theorem 1, there exists a (possibly random) subsequence $X_{n_k}$ of $X_n$ that converges to $\mathcal{X}^*$; formally, $\liminf_{n\to\infty} \operatorname{dist}(X_n, \mathcal{X}^*) = 0$ (a.s.).*

Before proving Lemma B.1, we will require an intermediate result:

**Lemma B.2.** *Let $\mathcal{C}$ be a closed subset of $\mathbb{R}^d$ such that $\mathcal{X}^* \cap \mathcal{C} = \varnothing$. Then, under Assumption 3, $\inf_{x \in \mathcal{C}} \|\nabla f(x)\| > 0$.*

*Proof.* Arguing by contradiction, assume there exists some sequence $x_n \in \mathcal{C}$ such that $\|\nabla f(x_n)\| \to 0$ as $n \to \infty$. If $x_n$ admits a subsequence converging to some limit point $\hat{x} \in \mathcal{C}$, then, by continuity

(recall that $f$ is assumed $C^d$), we would also have $\|\nabla f(\hat{x})\| = 0$. In turn, this would imply $\hat{x} \in \mathcal{X}^*$, contradicting the assumption that $\mathcal{C}$ is closed and disjoint from $\mathcal{X}^*$.

Therefore, to prove our claim, it suffices to examine the case where $x_n$ has no convergent subsequence, i.e., $\liminf_{n \to \infty} \|x_n\| = \infty$. However, this would mean that the gradient sublevel set $M_\varepsilon = \{x \in \mathbb{R}^d : \|\nabla f(x)\| \le \varepsilon\}$ is unbounded for all $\varepsilon > 0$, in contradiction to Assumption 3. We conclude that $\liminf_{n \to \infty} \|\nabla f(x_n)\| > 0$ for every sequence $x_n$ in $\mathcal{C}$, i.e., $\liminf_{x \in \mathcal{C}} \|\nabla f(x)\| > 0$. ∎

*Proof of Lemma B.1.* Assume ad absurdum that the event

$$\Omega_0 = \{\liminf_{n \to \infty} \operatorname{dist}(X_n, \mathcal{X}^*) > 0\} \tag{B.6}$$

occurs with positive probability. By Lemma B.2, if $\liminf_{n \to \infty} \operatorname{dist}(X_n, \mathcal{X}^*) > 0$, we must also have $\liminf_{n \to \infty} \|\nabla f(X_n)\| > 0$ (since $X_n$ will eventually be contained in a closed set that is disjoint from $\mathcal{X}^*$). Therefore, fixing a realization $X_n$, $n = 1, 2, \ldots$, of (SGD) such that $\Omega_0$ holds, there exists some (random) positive constant $c > 0$ with $\|f(X_n)\|^2 \ge c$ for all sufficiently large $n$; without loss of generality, we may – and will – assume in the sequel that this actually holds for all $n \ge 1$.

In view of all this, by the smoothness assumption for $f$ and the definition of (SGD) we readily get:

$$
\begin{aligned}
f(X_{n+1}) = f(X_n - \gamma_n V_n) &\le f(X_n) - \gamma_n \langle \nabla f(X_n), V_n \rangle + \frac{L}{2} \gamma_n^2 \|V_n\|^2 \\
&= f(X_n) - \gamma_n \|\nabla f(X_n)\|^2 - \gamma_n \langle \nabla f(X_n), Z_n \rangle + \frac{L}{2} \gamma_n^2 \|V_n\|^2 \\
&\le f(X_n) - \gamma_n c - \gamma_n \xi_n + \gamma_n^2 L \|\nabla f(X_n)\|^2 + \gamma_n^2 L \|Z_n\|^2, \quad \text{(B.7)}
\end{aligned}
$$

where we set $\xi_n = \langle \nabla f(X_n), Z_n \rangle$. Therefore, setting $f_n = f(X_n)$ and telescoping, we obtain

$$f_{n+1} \le f_1 - \tau_n \left[ c + \underbrace{\frac{\sum_{k=1}^n \gamma_k \xi_k}{\tau_n}}_{A_n} - L \underbrace{\frac{\sum_{k=1}^n \gamma_k^2 \|\nabla f(X_k)\|^2}{\tau_n}}_{B_n} - L \underbrace{\frac{\sum_{k=1}^n \gamma_k^2 \|Z_k\|^2}{\tau_n}}_{C_n} \right], \tag{B.8}$$

where $\tau_n = \sum_{k=1}^n \gamma_k$ is the "elapsed time" of $X_n$ as defined in Section 3. We will proceed to show that all the summands in the brackets of (B.8) except the first converge to 0; since $c > 0$ and $\tau_n \uparrow \infty$, this will show that $\lim_{n \to \infty} f_n = -\infty$, in direct contradiction to Assumption 2.

We carry out this plan term-by-term below:
1. For the first term $(A_n)$, note that $\mathbb{E}[\xi_n \,|\, \mathcal{F}_n] = \mathbb{E}[\langle \nabla f(X_n), Z_n \rangle \,|\, \mathcal{F}_n] = \langle \nabla f(X_n), \mathbb{E}[Z_n \,|\, \mathcal{F}_n] \rangle = 0$ by Assumption 4. This means that $\sum_{k=1}^n \gamma_k \xi_k$ is a zero-mean martingale, so, by the law of large numbers for martingale difference sequences [18, Theorem 2.18], we have $\gamma_n^{-1} \sum_{k=1}^n \gamma_k \xi_k \to 0$ (a.s.) on the event

$$\Omega_1 = \left\{ \sum_{n=1}^\infty \frac{\gamma_n^2}{\tau_n^2} \mathbb{E}[\xi_n^2 \,|\, \mathcal{F}_n] < \infty \right\}. \tag{B.9}$$

However, by Assumptions 1 and 4, we have

$$
\begin{aligned}
\mathbb{E}[\xi_n^2 \,|\, \mathcal{F}_n] &= \mathbb{E}[\langle \nabla f(X_n), Z_n \rangle^2 \,|\, \mathcal{F}_n] \\
&\le G^2 \, \mathbb{E}[\|Z_n\|^2 \,|\, \mathcal{F}_n] && \text{\{by Assumption 1\}} \\
&\le G^2 \, \mathbb{E}[\|Z_n\|^q \,|\, \mathcal{F}_n]^{2/q} && \text{\{by Jensen\}} \\
&\le G^2 \sigma^2 && \text{\{by Assumption 4\}}
\end{aligned}
$$

where, in the second-to-last line, we applied Jensen's inequality to the function $z \mapsto z^{q/2}$ (recall here that $q \ge 2$). Moreover, for all $p \in (0, 1]$, we have $\gamma_n^2 / \tau_n^2 = \tilde{\mathcal{O}}(1/n^2)$, so $\sum_{n=1}^\infty \gamma_n^2 / \tau_n^2 < \infty$. Thus, going back to (B.9), we conclude that $\gamma_n^{-1} \sum_{k=1}^n \gamma_k \xi_k \to 0$ with probability 1.

2. For the second term $(B_n)$, simply note that $\|\nabla f(X_n)\|^2 \leq G^2$, so we have:

$$B_n = \frac{\sum_{k=1}^n \gamma_k^2 \|\nabla f(X_k)\|^2}{\tau_n} \leq G^2 \frac{\sum_{k=1}^n \gamma_k^2}{\tau_n} = \begin{cases} \mathcal{O}(1/n^p) & \text{if } 0 < p < 1/2, \\ \mathcal{O}(\log n/\sqrt{n}) & \text{if } p = 1/2, \\ \mathcal{O}(1/n^{1-p}) & \text{if } 1/2 < p < 1, \\ \mathcal{O}(1/\log n) & \text{if } p = 1. \end{cases} \tag{B.10}$$

Thus, from the above, we conclude that $B_n \to 0$.

3. For the third term $(C_n)$, we will require a series of estimates. First, with a fair degree of hindsight, let $Q_n = \sum_{k=1}^n \gamma_k^{1+q/2} \|Z_k\|^q$. Noting that $\mathbb{E}[\|Z_n\|^q] < \infty$ (a.s.) and $\mathbb{E}[Q_n \mid \mathcal{F}_n] = Q_{n-1} + \gamma_n^{1+q/2} \|Z_n\|^q \geq Q_{n-1}$ for all $n = 1, 2, \ldots$, we deduce that $Q_n$ is a submartingale. Furthermore, we have:

$$\mathbb{E}\left[\sum_{n=1}^\infty \gamma_n^{1+q/2} \|Z_n\|^q\right] \leq \sum_{n=1}^\infty \gamma_n^{1+q/2} \mathbb{E}[\|Z_n\|^q] \leq \sum_{n=1}^\infty \gamma_n^{1+q/2} \sigma^q$$
$$= \mathcal{O}\left(\sum_{n=1}^\infty n^{-\frac{p(q+2)}{2}}\right) < \infty, \tag{B.11}$$

i.e., $Q_n$ is bounded in $L^1$ (recall that $p > 2/(q+2)$ by assumption). Hence, by Doob's submartingale convergence theorem [18, Theorem 2.1], it follows that $Q_n$ converges (a.s.) to a random variable $Q_\infty$ with $\mathbb{E}[Q_\infty] < \infty$ (and hence $Q_\infty < \infty$ with probability 1 as well).

To proceed, we will need to consider two cases, depending on whether $q = 2$ or $q > 2$. For the latter (which is more difficult), we will require the following variant of Hölder's inequality:

$$\left(\sum_{k=1}^n \alpha_k \beta_k\right)^r \leq \left(\sum_{k=1}^n \alpha_k^{\frac{\delta r}{r-1}}\right)^{r-1} \sum_{k=1}^n \alpha_k^{(1-\delta)r} \beta_k^r, \tag{B.12}$$

valid for all $r > 1$ and all $\delta \in (0, 1)$. Then, applying this inequality with $\alpha_k = \gamma_k^2$, $\beta_k = \|Z_k\|^2$, $r = q/2$ and $\delta = (q-2)/(2q)$, we obtain:

$$\left(\sum_{k=1}^n \gamma_k^2 \|Z_k\|^2\right)^{q/2} \leq \left(\sum_{k=1}^n \gamma_k\right)^{q/2-1} \sum_{k=1}^n \gamma_k^{1+q/2} \|Z_k\|^q = \tau_n^{q/2-1} Q_n, \tag{B.13}$$

and hence:

$$C_n = \frac{\sum_{k=1}^n \gamma_k^2 \|Z_k\|^2}{\tau_n} \leq \frac{\tau_n^{1-2/q} Q_n^{2/q}}{\tau_n} = \frac{Q_n^{2/q}}{\tau_n^{2/q}}. \tag{B.14}$$

Since $q > 2$ and $Q_n$ converges (a.s.) to $Q_\infty$, it follows that $\lim_{n\to\infty} C_n = 0$ with probability 1 (since $\lim_{n\to\infty} \tau_n = \infty$ by our assumptions for $\gamma_n$). Finally, if $q = 2$, we have $C_n = Q_n/\tau_n$ by definition, so we get $C_n \to 0$ (a.s.) directly.

Putting together all of the above, we get $A_n + B_n + C_n \to 0$ with probability 1, and hence, with probability 1 conditioned on $\Omega_0$ (since $\mathbb{P}(\Omega_0) > 0$). This means that, for sufficiently large $n$, we have

$$f_{n+1} \leq f_1 - \tau_n(c/2) \tag{B.15}$$

which, together with the fact that $\lim_{n\to\infty} \tau_n = \infty$, implies that $\lim_{n\to\infty} f(X_n) = -\infty$. This contradicts Assumption 2 and completes our proof. ∎

We now move on to the deterministic elements of the proof of Theorem 1. To that end, let

$$a = \max_{x \in \mathcal{X}^*} f(x) \tag{B.16}$$

denote the maximum value of $f$ over its critical set, and let

$$K_\varepsilon = L_{a+\varepsilon} = \{x \in \mathbb{R}^d : f(x) \leq a + \varepsilon\} \tag{B.17}$$

denote the $(a+\varepsilon)$-sublevel set of $f$ (which is bounded by the fact that $f$ is coercive, cf. Assumption 2). We then have the following "uniform decrease" estimate:

**Lemma B.3.** *Fix some $\varepsilon > 0$. Under Assumptions 1–3, there exists some $\tau \equiv \tau(\varepsilon)$ such that, for all $x \in \mathbb{R}^d$, we have (i) $f(\Phi_\tau(x)) \leq f(x) - \varepsilon$; or (ii) $\Phi_\tau(x) \in K_\varepsilon$.*

*Proof.* By Lemma B.2, there exists some positive constant $c > 0$ such that $\|\nabla f(x)\|^2 \geq c > 0$ for all $x \in \mathbb{R}^d \setminus K_\varepsilon$. Then, with $df/dt = -\|\nabla f(x(t))\|^2$, if we let $\tau_x = \inf\{t \geq 0 : \Phi_t(x) \in K_\varepsilon\}$, we get:

$$f(\Phi_t(x)) = f(x) - \int_0^t \|\nabla f(x(s))\|^2 \, ds \leq f(x) - ct \quad \text{for all } t \in [0, \tau_x]. \tag{B.18}$$

Accordingly, letting $\tau = \varepsilon/c$, we may consider the following two case:

1. If $\tau_x \geq \tau$, applying (B.18) for $t = \tau$ yields $f(\Phi_\tau(x)) \leq f(x) - \varepsilon$.

2. Otherwise, if $\tau_x < \tau$, we have $f(\Phi_\tau(x)) \leq f(\Phi_{\tau_x}(x)) \leq a + \varepsilon$, implying in particular that $\Phi_\tau(x) \in K_\varepsilon$.

Our claim then follows by combining the two cases above. ∎

Finally, we establish below the required comparison bound between an APT of (GD) and its solution trajectories:

**Lemma B.4.** *Fix some $\varepsilon, \delta > 0$. Then, with assumptions and notation as in Lemma B.3, there exists some $t_0 \equiv t_0(\delta, \varepsilon)$ such that, for all $t \geq t_0$ and all $h \in [0, \tau]$, we have:*

$$f(X(t+h)) \leq f(\Phi_h(X(t))) + G\delta + \tfrac{1}{2}L\delta^2. \tag{B.19}$$

*Proof.* By the definition of an APT, there exists some $t_0 \equiv t_0(\delta, \varepsilon)$ such that

$$\sup_{0 \leq h \leq \tau} \|X(t+h) - \Phi_h(X(t))\| \leq \delta \tag{B.20}$$

for all $t \geq t_0$. Hence, for all $t \geq t_0$ and all $h \in [0, \tau]$, we have

$$\begin{aligned}
f(X(t+h)) &= f(\Phi_h(X(t)) + X(t+h) - \Phi_h(X(t))) \\
&\leq f(\Phi_h(X(t))) + \langle \nabla f(\Phi_h(X(t))), X(t+h) - \Phi_h(X(t)) \rangle \\
&\quad + \frac{L}{2}\|X(t+h) - \Phi_h(X(t))\|^2 \\
&\leq f(\Phi_h(X(t))) + G\|X(t+h) - \Phi_h(X(t))\| + \frac{L}{2}\|X(t+h) - \Phi_h(X(t))\|^2 \\
&\leq f(\Phi_h(X(t))) + G\delta + \frac{L}{2}\delta^2,
\end{aligned} \tag{B.21}$$

as claimed. ∎

With all this in hand, we are finally in a position to formally prove Theorem 1.

*Proof of Theorem 1.* We will prove the stronger statement that, with probability 1, $X(t)$ converges to the sublevel set $L_a = \{x \in \mathbb{R}^d : f(x) \leq a\}$ with $a$ defined as in (B.16). Since $f$ is coercive (cf. Assumption 2), the sublevel sets of $f$ are bounded; hence, proving convergence to $L_a$ suffices.

To prove this claim, fix some $\varepsilon > 0$ and let $X(t)$ be the affine interpolation of the sequence of iterates $X_n$ generated by (SGD). Under the stated assumptions, Proposition 1 guarantees that $X(t)$ is an APT of (GD) with probability 1. Moreover, again with probability 1, Lemma B.1 guarantees the existence of some (possibly random) $t_1$ such that $X(t_1) \in K_{2\varepsilon}$. To streamline the analysis to come, we will condition our statements on the intersection of these two events (which still occurs with probability 1), and we will argue trajectory-wise.

Moving forward, Lemma B.3 guarantees the existence of some $\tau \equiv \tau(\varepsilon)$ such that $f(\Phi_\tau(x)) \leq f(x) - \varepsilon$ or $\Phi_\tau(x) \in K_\varepsilon$ for all $x \in \mathbb{R}^d$. Fixing this $\tau$ and taking $\delta > 0$ such that $G\delta + L\delta^2/2 < \varepsilon$, Lemma B.4 further implies that there exists some $t_0$ such that (B.19) holds for all $t \geq t_0$ and all $h \in [0, \tau]$. Note also that, without loss of generality, we can assume that $t_1 > t_0$; otherwise, if this is not the case, it suffices to wait for the first instance $n$ such that $X_n \in K_{2\varepsilon}$ and $\tau_n \geq t_0$ (by Lemma B.1, this occurs with probability 1).

Combining all of the above, we have (*i*) $X(t_1) \in K_{2\varepsilon}$; and (*ii*) $f(X(t+h)) \leq f(\Phi_h(X(t))) + \varepsilon$ for all $t \geq t_1$ and all $h \in [0, \tau]$. Since $f(\Phi_t(x)) \leq f(x)$ for all $t \geq 0$, this further implies that

$$f(X(t+h)) \leq f(X(t)) + \varepsilon \tag{B.22}$$

for all $h \in [0, \tau]$. We thus get

$$f(X(t)) \leq f(X(t_1)) + \varepsilon \leq a + 3\varepsilon \tag{B.23}$$

for all $t \in [t_1, t_1 + \tau]$. Moreover, since $X(t_1) \in K_{2\varepsilon}$, Lemma B.3 also gives $\Phi_\tau(X(t_1)) \in K_\varepsilon$ because the two conditions of the lemma coincide if $x \in K_{2\varepsilon}$. As a result, we finally obtain

$$f(X(t_1 + \tau)) \leq f(\Phi_\tau(X(t_1))) + \varepsilon \leq a + \varepsilon + \varepsilon = a + 2\varepsilon, \tag{B.24}$$

i.e., $X(t_1 + \tau) \in K_{2\varepsilon}$.

From the above, we conclude that (*i*) $X(t) \in K_{3\varepsilon}$ for all $t \in [t_1, t_1 + \tau]$; and, in particular, (*ii*) $X(t_1 + \tau) \in K_{2\varepsilon}$. Proceeding inductively, we get $X(t) \in K_{3\varepsilon}$ for all $t \in [t_1 + (k-1)\tau, t_1 + k\tau]$, $k = 1, 2, \ldots$, i.e., $X(t) \in K_{3\varepsilon}$ for all $t \geq t_1$. Since $\varepsilon > 0$ is arbitrary, this means that $X(t)$ converges to $K_0 \equiv L_a$ as claimed. ∎

# C Avoidance analysis

As we stated in the main body of the paper, the proof of Theorem 3 will require two different threads of arguments: *a*) a series of probabilistic estimates to show that a certain class of stochastic processes avoids zero; and *b*) the construction of a suitable (average) Lyapunov function that grows exponentially along the unstable directions of a strict saddle manifold.

**C.1. Probabilistic estimates.** The probabilistic estimates that we will need date back to Pemantle [40] and concern a class of stochastic processes defined as follows: let $Y_n$, $n = 1, 2, \ldots$, be a sequence of $\mathcal{F}_n$-measurable random variables, let $E_n = \sum_{k=1}^n Y_k$, and assume that

$$\mathbb{E}[E_{n+1}^2 - E_n^2 \,|\, \mathcal{F}_n] \geq C/n^{2p} \quad \text{for some } C > 0 \text{ and all } n = 1, 2, \ldots \tag{C.1}$$

In the above, $E_n$ will play the role of a "distance measure" from $\mathcal{S}$. Informally, the requirement (C.1) posits that $E_n$ increases in "root mean square" by $\Theta(\gamma_n)$ where $\gamma_n \propto 1/n^p$ is the step-size of (SGD); constructing such a process will be the topic of the geometric constructions of the next section. For now, we state without proof a number of conditions guaranteeing that the process $E_n$ cannot converge to $0$:

**Lemma C.1** ($0 < p \leq 1/2$; 4, Lemma 4.2). *Suppose that* (C.1) *holds for some* $p \in (0, 1/2]$. *Then,* $\mathbb{P}(\lim_{n\to\infty} E_n = 0) = 0$.

**Lemma C.2** ($1/2 < p \leq 1$; 41, Lemma 5.5). *Suppose that* (C.1) *holds for some* $p \in (1/2, 1]$. *Assume further that there exist constants* $a, b > 0$ *such that, for all* $n = 1, 2, \ldots$, *we have:*

1. $|Y_n| \leq a/n^p$ *with probability* $1$.

2. $\mathbb{1}_{\{E_n > b/n^p\}} \mathbb{E}[Y_{n+1} \,|\, \mathcal{F}_n] \geq 0$ *with probability* $1$.

*Then,* $\mathbb{P}(\lim_{n\to\infty} E_n = 0) = 0$.

A first version of Lemma C.2 was originally proven by Pemantle [40] for the special case $p = 1$ but the proof techniques are similar for all $1/2 < p \leq 1$; for a more general estimate (which we will not need here), see Benaïm [2, Lemma 9.6].

**C.2. Center manifold theory and geometric constructions.** We now proceed with the construction of a suitable Lyapunov function that will allow us to apply Lemmas C.1 and C.2. This construction follows Benaïm and Hirsch [3] and Benaïm [2] and relies crucially on center manifold theory; for a general introduction to the topic, we refer the reader to Lee [29], Shub [46], and Robinson [45].

To begin, let $\mathcal{S}$ be a strict saddle manifold as defined in Section 4.2. Then, for all $x^* \in \mathcal{S}$, we define the *center*, *stable* and *unstable* directions of $x^*$ to be respectively the eigenspaces of $H(x^*) = \nabla^2 f(x^*)$ corresponding to zero, positive and negative eigenvalues thereof, i.e.,

$$\mathcal{E}_{x^*}^c = \{v \in \mathbb{R}^d : H(x^*)v = 0\} = \ker H(x^*), \qquad \text{[central directions]} \tag{C.2a}$$

$$\mathcal{E}_{x^*}^s = \{v \in \mathbb{R}^d : H(x^*)v = \lambda v \text{ for some } \lambda > 0\} \qquad \text{[stable directions]} \tag{C.2b}$$

$$\mathcal{E}_{x^*}^u = \{v \in \mathbb{R}^d : H(x^*)v = \lambda v \text{ for some } \lambda < 0\} \qquad \text{[unstable directions]} \tag{C.2c}$$

The reason for this terminology is that $H(x) = \mathrm{Jac}(\nabla f(x))$, so these subspaces correspond to directions that are respectively neutral (or *slow*), attracting, and repelling under (GD). More precisely, by the center manifold theorem [45, 46], there exists a neighborhood $\mathcal{U}$ of $\mathcal{S}$ and a submanifold $\mathcal{M}$ of $\mathbb{R}^d$, called the *center stable manifold* of $\mathcal{S}$, and satisfying the following: *a)* $\mathcal{M}$ is *locally invariant* under $\Phi$, i.e., there exists some positive $t_0 > 0$ such that $\Phi_t(\mathcal{U} \cup \mathcal{M}) \subseteq \mathcal{M}$ for all $t \geq t_0$; and *b)* $\mathbb{R}^d = T_{x^*}\mathcal{M} \oplus \mathcal{E}_{x^*}^u$ for all $x^* \in \mathcal{S}$, where $T_{x^*}\mathcal{M}$ denotes the tangent space to $\mathcal{M}$ at $x^*$. In view of this: *a)* perturbations along central directions are tangent to $\mathcal{M}$ and are thus expected to evolve "along" $\mathcal{M}$ under (GD); *b)* stable perturbations along $\mathcal{E}_{x^*}^s$ will converge along $\mathcal{M}$ to $\mathcal{S}$ under (GD); and *c)* unstable perturbations along $\mathcal{E}_{x^*}^u$ are transverse to $\mathcal{M}$ and may escape.

A key property of $\mathcal{M}$ is that any globally bounded orbit of (GD) which is contained in a sufficiently small neighborhood of $x^* \in \mathcal{S}$ must be entirely contained in $\mathcal{M}$ [46]. Moreover, by the non-minimality assumption for $\mathcal{S}$, it follows that $d_u \equiv \dim \mathcal{E}_{x^*}^u \geq 1$, so the dimension of $\mathcal{M}$ is at most $d - 1$. This suggests that perturbations along any direction that is transverse to $\mathcal{M}$ will be repelled under (GD); we make this statement precise in the lemma below.

**Lemma C.3.** *Let $\Psi_t(x) = \nabla_x \Phi_t(x^*)$ denote the infinitesimal generator of the flow of (GD). Then:*

1. *The unstable subspaces $\mathcal{E}_{x^*}^u$ are invariant under (GD); specifically, $\Psi_t(x^*)\mathcal{E}_{x^*}^u = \mathcal{E}_{x^*}^u$ for all $t \geq 0$ and all $x^* \in \mathcal{S}$.*

2. *There exists a positive constant $c > 0$ such that, for all $x^* \in \mathcal{S}$, $w \in \mathcal{E}_{x^*}^u$ and $t \geq 0$, we have*

$$\|\Psi_t(x^*)w\| \geq e^{ct}\|w\|. \tag{C.3}$$

*Remark* 3. In the above (and what follows), we write $AW$ for the image of a vector space $W$ under a linear operator $A$. Specifically, if $A: V \to V'$ is a linear operator between two vector spaces $V$ and $V'$, and if $W \leq V$ is a subpace of $V$, we let $AW \equiv \mathrm{im}_A(W) = \{Aw : w \in W\}$. We also treat linear operators and matrices interchangeably.

*Remark* 4. The proof of Lemma C.3 (and, in fact, all of our analysis in this section) does not require the uniformity condition $\min \lambda_+(H(x^*)) \geq c_+$ for the Hessian's positive eigenvalues (if such eigenvalues exist). We only make it to simplify the presentation and avoid cases where the dimension of $\mathcal{E}_{x^*}^s$ may change; in that case, it would be sufficient to work with a subset of $\mathcal{S}$ over which this does not occur.

In words, Lemma C.3 states that *a)* the unstable directions along $\mathcal{S}$ are consistent with the flow of (GD); and *b)* perturbations along unstable directions are repelled from $\mathcal{S}$ at a geometric rate. The proof is as follows:

*Proof of Lemma C.3.* Recall first that, for all $t \geq 0$ and all $x \in \mathbb{R}^d$, we have $\Psi_t(x) = \nabla_x \Phi_t(x) = \exp(t \, \mathrm{Jac}(-\nabla f(x))) = \exp(-tH(x))$. Therefore, since $\mathcal{S}$ consists entirely of stationary points of (GD), we readily get

$$\begin{aligned}
\Psi_t(x^*)\mathcal{E}_{x^*}^u &= e^{-tH(x^*)}\mathcal{E}_{x^*}^u \\
&= \sum_{k=0}^{\infty} \frac{(-t)^k}{k!} H(x^*)^k \mathcal{E}_{x^*}^u = \sum_{k=0}^{\infty} \frac{(-t)^k}{k!} \mathcal{E}_{x^*}^u \qquad \{\text{because } H(x^*)\mathcal{E}_{x^*}^u = \mathcal{E}_{x^*}^u\} \\
&= e^{-t}\mathcal{E}_{x^*}^u = \mathcal{E}_{x^*}^u, \qquad\qquad\qquad\qquad\qquad\qquad\qquad\qquad\quad \{\text{C.4}\}
\end{aligned}$$

so our first claim follows.

For our second claim, let $\{u_i : i = 1, \ldots, d\}$ be an orthnormal set of eigenvectors of $H(x^*)$.[3] Moreover, let $\lambda_i \equiv \lambda_i(x^*) < 0$ be the eigenvalue of $H(x^*)$ corresponding to $u_i$, and assume without loss of generality that the indexing labels $i = 1, \ldots, d$ have been chosen in ascending eigenvalue order, i.e., $\lambda_1 \leq \cdots \leq \lambda_d$. It then follows that $\{u_i : i = 1, \ldots, d_u \equiv \dim \mathcal{E}_{x^*}^u\}$ is an orthonormal basis of $\mathcal{E}_{x^*}^u$ consisting entirely of eigenvectors of $H(x^*)$. Thus, writing $w = \sum_i w_i u_i$ for a given vector $w \in \mathcal{E}_{x^*}^u$, we have:

$$\Psi_t(x^*)w = e^{-tH(x^*)}w = \sum_{i=1}^{d_u} w_i e^{-tH(x^*)}u_i = \sum_{i=1}^{d_u} w_i e^{-t\lambda_i}u_i, \tag{C.5}$$

where, in the last step, we used the fact that $u_i$ is an eigenvector of $H(x^*)$ with eigenvalue $\lambda_i$ (and hence, also of $e^{-tH(x^*)}$ with eigenvalue $e^{-t\lambda_i}$). Therefore, by orthonormality, we obtain:

$$\|\Psi_t(x^*)w\|^2 = \sum_{i=1}^{d_u} e^{-2t\lambda_i} w_i^2 \geq e^{2c_- t} \|w\|^2, \tag{C.6}$$

where $c_- > 0$ is defined in Section 4.3. $\blacksquare$

To proceed, we will need to define a suitable "projector" from neighborhoods of $\mathcal{S}$ to $\mathcal{M}$. To carry out this construction, consider the vector bundle

$$\mathcal{E}_{\mathcal{S}}^u \equiv \{(x^*, w) : x^* \in \mathcal{S}, w \in \mathcal{E}_{x^*}^u\} \tag{C.7}$$

of the unstable directions of (GD) over $\mathcal{S}$. Since each $\mathcal{E}_{x^*}^u$ is a subspace of $\mathbb{R}^d$, we can view $\mathcal{E}_{\mathcal{S}}^u$ as a map from $\mathcal{S}$ to the Grassmannian $\mathbf{Gr}(d_u, d)$ of $d_u$-dimensional spaces of $\mathbb{R}^d$. By the Whitney embedding theorem [29], $\mathbf{Gr}(d_u, d)$ can be embedded as a $d_u \times (d - d_u)$-dimensional submanifold of $\mathbb{R}^{2d_u(d-d_u)}$; as such, $\mathcal{E}_{\mathcal{S}}^u$ may be seen as a map $\mathcal{S} \to \mathbb{R}^{2d_u(d-d_u)}$ with values in $\mathbf{Gr}(d_u, d) \hookrightarrow \mathbb{R}^{2d_u(d-d_u)}$. Since $\mathcal{S}$ is closed (as a connected component of $\mathcal{X}^*$), the Tietze extension theorem [1] further implies that this map admits a continuous extension $\pi \colon \mathbb{R}^d \to \mathbb{R}^{2d_u(d-d_u)}$ to all of $\mathbb{R}^d$. By mollifying this map with an approximate identity supported on $\mathcal{S}$, we can further assume that this extension is smooth in a neighborhood of $\mathcal{S}$. Moreover, by standard results in differential topology [19, Chap. 4], there exists a smooth retraction of a neighborhood of $\mathbf{Gr}(d_u, d)$ onto $\mathbf{Gr}(d_u, d)$ in $\mathbb{R}^{2d_u(d-d_u)}$. Hence, by composing $\pi$ with this retraction, we finally obtain a smooth vector bundle

$$\mathcal{E}_{\mathcal{U}}^u \equiv \{(x, w) : x \in \mathcal{U}, w \in \mathcal{E}_x^u\} \tag{C.8}$$

which, by construction, coincides with $\mathcal{E}_{\mathcal{S}}^u$ over $\mathcal{S}$ (explaining the slight abuse of notation).

By taking a smaller neighborhood if necessary, we may assume that $\mathcal{U}$ is compact and coincides with the one in the definition of $\mathcal{M}$, i.e., $\Phi_t(\mathcal{U} \cap \mathcal{M}) \subseteq \mathcal{M}$ for small enough $t$. We may now construct a "projector" from a (potentially smaller) neighborhood of $\mathcal{M}$ to $\mathcal{M}$ as follows: First, consider the simple vector addition mapping $Q \colon \mathcal{E}_{\mathcal{U}}^u \to \mathbb{R}^d \equiv \mathbb{R}^d$ sending $(x, w) \in \mathcal{E}_{\mathcal{U}}^u \mapsto x + w \in \mathbb{R}^d$. Clearly, the zero section $(x, 0)$ of $\mathcal{E}_{\mathcal{U}}^u$ is mapped diffeomorphically to $\mathcal{U}$ so, by the inverse function theorem [29], it follows that $Q$ is a local diffeomorphism. Thus, letting $\mathcal{U}'$ be a neighborhood of $\mathcal{M}$ over which $Q$ is a diffeomorphism, and letting $\mathcal{U}_0 = Q(\mathcal{U}')$, we get a map $\Pi \colon \mathcal{U}_0 \to \mathcal{M}$ such that

$$\Pi(y) = x \iff Q(x, w) = x + w = y \tag{C.9}$$

The reason for this sophisticated construction (as opposed to e.g., taking a Euclidean projection from $\mathcal{U}_0$ to $\mathcal{M}$) is that $\Pi$ respects the unstable directions of $\mathcal{S}$ under (GD). More precisely, we have:

**Lemma C.4.** *For $x \in \mathcal{U}$, let $P_x \colon T_x\mathcal{M} \oplus \mathcal{E}_x^u \to T_x\mathcal{M}$ denote the projection*

$$\underset{\substack{\cap \\ T_x\mathcal{M} \oplus \mathcal{E}_x^u}}{z + w} \mapsto P_x(z + w) = \underset{\substack{\cap \\ T_x\mathcal{M}}}{z} \tag{C.10}$$

*Then, for all $x \in \mathcal{U}_0 \cap \mathcal{M}$, we have $\mathrm{Jac}(\Pi(x)) = P_x$.*

*Proof.* Let $y(t)$, $t \in (-1, 1)$ be a smooth curve on $\mathcal{U}_0$ going through $x = y(0) \in \mathcal{M}$ at time $t = 0$, and let $x(t) = \Pi(y(t))$ so $y(t) = x(t) + \psi(t)$ for some smooth $\psi(t) \in \mathcal{E}_{x(t)}^u$. By differentiating, we get $\dot{y}(0) = \dot{x}(0) + \dot{\psi}(0)$; since $x(t) \in \mathcal{M}$ and $\psi(t) \in \mathcal{E}_{x(t)}^u$ for all $t$, we readily get $\dot{x}(0) \in T_{x(0)}\mathcal{M}$ and $\dot{\psi}(0) \in \mathcal{E}_{x(0)}^u$. Letting $z = \dot{x}(0)$ and $w = \dot{\psi}(0)$, this shows that the pushforward of $\dot{y}(0) = z + w$ under $\Pi$ at $x$ is $D\Pi_x(z + w) \equiv \mathrm{Jac}(\Pi(x))(z + w) = z = P_x(z + w)$. With $y(t)$ arbitrary, our claim follows. $\blacksquare$

We are finally in a position to define a "potential function" on $\mathcal{U}_0$ as

$$V(y) = \|\Pi(y) - y\| \tag{C.11}$$

i.e., as the (normed) distance of $y \in \mathcal{U}_0$ from its vector projection $\Pi(y)$ on $\mathcal{M}$ along the unstable directions of (GD). By construction, we have

$$V(y) \geq 0 \quad \text{with equality if and only if } y \in \mathcal{M} \cap \mathcal{U}_0. \tag{C.12}$$

Coupling (C.12) with Lemmas C.3 and C.4, we see that $f$ satisfies the requirements of Benaïm [2, Proposition 9.5], which, when adapted to our setting, provides the following:

**Proposition C.1** (2). *There exists a compact neighborhood $\mathcal{U}_{\mathcal{S}}$ of $\mathcal{S}$, a positive constant $\beta > 0$, and a time horizon $\tau > 0$ such that the energy function*

$$E(x) = \int_0^\tau V(\Phi_{-t}(x))\, dt \qquad x \in \mathcal{U}_{\mathcal{S}}, \tag{C.13}$$

*enjoys the following properties:*

1. *For all $x \in \mathcal{U}_{\mathcal{S}}$, $E$ has a Lipschitz continuous and positively homogeneous right derivative $\nabla^+ E(x)$;[4] in addition, $E$ is continuously differentiable on $\mathcal{U}_{\mathcal{S}} \setminus \mathcal{M}$.*

2. *For all $x \in \mathcal{U}_{\mathcal{S}}$, we have*
$$\nabla^+ E(x)[\nabla f(x)] \leq -\beta E(x). \tag{C.14}$$
   *In particular, for all $x \in \mathcal{U}_{\mathcal{S}} \setminus \mathcal{M}$, we have:*
$$\langle \nabla E(x), \nabla f(x) \rangle \leq -\beta E(x) \tag{C.15}$$

3. *There exists a constant $\alpha > 0$ such that, for all $x \in \mathcal{U}_{\mathcal{S}}$ and all sufficiently small $v \in \mathbb{R}^d$, we have*
$$E(x + v) \geq E(x) + \nabla^+ E(x)[v] - \frac{\alpha}{2}\|v\|^2. \tag{C.16}$$

4. *There exists a constant $\beta > 0$ such that, for all $v \in \mathbb{R}^d$, we have:*
$$\|\nabla E(x)\| \geq \beta \qquad \text{for all } x \in \mathcal{U}_{\mathcal{S}} \setminus \mathcal{M}, \tag{C.17a}$$
   *and*
$$\nabla^+ E(x)[v] \geq \beta \|P_x(v) - v\| \qquad \text{for all } x \in \mathcal{U}_{\mathcal{S}} \cap \mathcal{M}. \tag{C.17b}$$

Proposition C.1 follows from Benaïm [2, Proposition 9.5], so we do not present a proof. More important for our purposes are the following immediate consequences thereof:

1. By (C.15), the energy $E(x(t))$ of a solution orbit $x(t)$ of (GD) will grow at a (locally) geometric rate if $x(t)$ doesn't already lie in the center stable manifold $\mathcal{M}$ of $\mathcal{S}$. This means that asymptotic pseudotrajectories of (GD) that do not lie on $\mathcal{M}$ for arbitrarily long windows of time will also escape $\mathcal{M}$ (and hence $\mathcal{S}$).

2. The bound (C.16) provides the basis for a discrete-time version of the above argument: as long as $X_n$ is sufficiently close to $\mathcal{S}$, the energy before and after a stochastic gradient step will be linked as

$$E(X_{n+1}) \geq E(X_n) + \beta\gamma_n E(X_n) - \gamma_n \psi_n - \frac{\alpha\gamma_n^2}{2}\|V_n\|^2, \tag{C.18}$$

   where $\psi_n$ is an additive noise term which is non-antagonistic in expectation. This means that, on average, the iterates $E_n \equiv E(X_n)$ will grow at a (locally) geometric rate, so $X_n$ cannot remain in the vicinity of $\mathcal{S}$ for very long periods.

To make the above precise, we will need to invoke the probabilistic estimates stated in Appendix C.1. We do so in the following section.

## C.3. Avoidance of saddle-point manifolds.   For convenience, we begin by restating our main avoidance result below:

**Theorem 3.** *Suppose that* (SGD) *is run with a variable step-size sequence of the form $\gamma_n \propto 1/n^p$ for some $p \in (0, 1]$. If Assumptions 1–5 hold (with $q = \infty$ for Assumption 4), and $\mathcal{S}$ is a strict saddle manifold of $f$, we have $\mathbb{P}(X_n \to \mathcal{S} \text{ as } n \to \infty) = 0$.*

*Proof.* Our proof follows the arguments of Benaïm and Hirsch [3], suitably adapted to our setting. To begin, let $\mathcal{U}_{\mathcal{S}}$ be the compact neighborhood of $\mathcal{S}$ identified in Proposition C.1 and assume without loss of generality that $X_1 \in \mathcal{U}_{\mathcal{S}}$. We may then define the exit time from $\mathcal{U}_{\mathcal{S}}$ as

$$T_{\mathcal{S}} = \inf\{n \geq 1 : n \notin \mathcal{U}_{\mathcal{S}}\}. \tag{C.19}$$

We will prove our claim by showing that $T_{\mathcal{S}} < \infty$ with probability 1.

To that end, consider the process

$$Y_{n+1} = \begin{cases} E(X_{n+1}) - E(X_n) & \text{if } n \leq T_{\mathcal{S}}, \\ \gamma_n & \text{otherwise,} \end{cases} \tag{C.20}$$

with $E(X_0) \equiv 0$ by convention. Heuristically, $Y_n$ measures the change in energy of $X_n$ as long as it remains in $\mathcal{U}_{\mathcal{S}}$; subsequently, for book-keeping purposes, it is incremented by a token amount of $\gamma_n$ per iteration once $X_n$ exits $\mathcal{U}_{\mathcal{S}}$. To make this idea more formal, let

$$E_n = \sum_{k=1}^{n} Y_k \tag{C.21}$$

so $E_n = E(X_n)$ if $X_n \in \mathcal{U}_{\mathcal{S}}$ while $E_n = \Theta(\tau_n)$ after $X_n$ exits $\mathcal{U}_{\mathcal{S}}$.

Assume now that $X_n \in \mathcal{U}_{\mathcal{S}}$ for all $n$ (i.e., $T_{\mathcal{S}} = \infty$). By Theorem 2, every limit point $\hat{x}$ of $X_n$ must be contained in $\mathcal{S}$, so, by (C.12), we must have $\lim_{n \to \infty} E_n = 0$. Hence, to establish our claim, it suffices to show that $\mathbb{P}(E_n \to 0) = 0$. We will do this by showing that $Y_n$ defined as in (C.20) satisfies the requirements of Lemmas C.1 and C.2.

We begin with the conditions required by Lemma C.2 for the case $1/2 < p \leq 1$:

1. For the condition $|Y_n| = \mathcal{O}(1/n^p)$ of Lemma C.2, the claim is tautological if $n > T_{\mathcal{S}}$. Otherwise, if $n \leq T_{\mathcal{S}}$, note that

$$\|X_{n+1} - X_n\| = \gamma_n \|V_n\| \leq \gamma_n[\|\nabla f(X_n)\| + \|Z_n\|] \leq \gamma_n (G + \sigma) \tag{C.22}$$

by Assumptions 1 and 4 (recall here that we are taking $q = \infty$ in Assumption 4). Since $X_n \in \mathcal{U}_{\mathcal{S}}$ as long as $n \leq T_{\mathcal{S}}$, and given that $E$ is continuously differentiable on $\mathcal{U}_{\mathcal{S}}$ (and hence Lipschitz continuous therein), we will also have:

$$|Y_{n+1}| = |E(X_{n+1}) - E(X_n)| = \mathcal{O}(\|X_{n+1} - X_n\|) = \mathcal{O}(\gamma_n) = \mathcal{O}(1/n^p), \quad \text{(C.23)}$$

as claimed.

2. For the condition $\mathbb{1}_{\{E_n > b/n^p\}} \mathbb{E}[Y_{n+1} \mid \mathcal{F}_n] \geq 0$, note first that if $n > T_{\mathcal{S}}$, then $Y_n = \gamma_n$, so

$$\mathbb{1}_{\{n > T_{\mathcal{S}}\}} \mathbb{E}[Y_{n+1} \mid \mathcal{F}_n] \geq \mathbb{1}_{\{n > T_{\mathcal{S}}\}} \gamma_n > 0. \tag{C.24}$$

Otherwise, if $n \leq T_{\mathcal{S}}$, we have $X_n \in \mathcal{U}_{\mathcal{S}}$, so Proposition C.1 yields

$$Y_{n+1} = E(X_{n+1}) - E(X_n) \geq \beta \gamma_n E(X_n) - \gamma_n \psi_n - 2\alpha \gamma_n^2 (G^2 + \sigma^2), \tag{C.25}$$

where we set

$$\psi_n = \nabla^+ E(X_n)[Z_n] \tag{C.26}$$

and used the estimate $\|V_n\|^2 = \|\nabla f(X_n) + Z_n\|^2 \leq [\|\nabla f(X_n)\|^2 + \|Z_n\|^2] \leq 2(G^2 + \sigma^2)$ (compare also with (C.18) and the surrounding discussion). By the conditional Jensen inequality and the definition of $\nabla^+ E(x)$, we have

$$\mathbb{E}[\psi_n \mid \mathcal{F}_n] = \mathbb{E}[\nabla^+ E(X_n)[Z_n] \mid \mathcal{F}_n] \geq \nabla^+ E(X_n)[\mathbb{E}[Z_n \mid \mathcal{F}_n]] = 0. \tag{C.27}$$

which, in turn, implies that

$$\mathbb{1}_{\{n \leq T_{\mathcal{S}}\}} \mathbb{E}[Y_{n+1} \mid \mathcal{F}_n] \geq \gamma_n \mathbb{1}_{\{n \leq T_{\mathcal{S}}\}} [\beta E(X_n) - 2\alpha (G^2 + \sigma^2) \gamma_n]. \tag{C.28}$$

Hence, taking $b > 0$ such that $b\beta/n^p = 2\alpha(G^2 + \sigma^2)\gamma_n$, and recalling that $E_n = E(X_n)$ if $n \leq T_{\mathcal{S}}$, we get

$$\begin{aligned} \mathbb{1}_{\{E_n > b/n^p\}} \mathbb{1}_{\{n \leq T_{\mathcal{S}}\}} \mathbb{E}[Y_{n+1} \mid \mathcal{F}_n] &\geq \mathbb{1}_{\{E_n > b/n^p \wedge n \leq T_{\mathcal{S}}\}} [\beta E_n - 2\alpha(G^2 + \sigma^2)\gamma_n] \\ &\geq \mathbb{1}_{\{E_n > b/n^p \wedge n \leq T_{\mathcal{S}}\}} [b\beta/n^p - 2\alpha(G^2 + \sigma^2)\gamma_n] \\ &\geq 0. \end{aligned} \tag{C.29}$$

Thus, combining the above, we conclude that the specific conditions required to apply Lemma C.2 are satisfied.

We are left to establish the general condition (C.1) which is required to apply both Lemmas C.1 and C.2; the proof is the same for all $p \in (0, 1]$, so we no longer assume $1/2 < p \leq 1$ below. To begin, note that

$$
\begin{aligned}
\mathbb{E}[E_{n+1}^2 - E_n^2 \,|\, \mathcal{F}_n] &= \mathbb{E}[Y_{n+1}^2 \,|\, \mathcal{F}_n] + 2E_n \,\mathbb{E}[Y_{n+1} \,|\, \mathcal{F}_n] \\
&= \mathbb{E}[Y_{n+1}^2 \,|\, \mathcal{F}_n] + 2E_n \,\mathbb{1}_{\{E_n \leq b/n^p\}} \,\mathbb{E}[Y_{n+1} \,|\, \mathcal{F}_n] \\
&\quad\quad + 2E_n \,\mathbb{1}_{\{E_n > b/n^p\}} \,\mathbb{E}[Y_{n+1} \,|\, \mathcal{F}_n] \\
&\geq \mathbb{E}[Y_{n+1}^2 \,|\, \mathcal{F}_n] + 2E_n \,\mathbb{1}_{\{E_n \leq b/n^p\}} \,\mathbb{E}[Y_{n+1} \,|\, \mathcal{F}_n], \quad\quad \text{(C.30)}
\end{aligned}
$$

where, in the last line, we used the inequalities proved in the previous paragraph, namely (C.24) and (C.29). To proceed, recall that $Y_n = \gamma_n > 0$ if $n > T_{\mathcal{S}}$, so, by (C.28) we get

$$
\begin{aligned}
\mathbb{E}[E_{n+1}^2 - E_n^2 \,|\, \mathcal{F}_n] &\geq \mathbb{E}[Y_{n+1}^2 \,|\, \mathcal{F}_n] \\
&\quad\quad + 2E_n \,\mathbb{1}_{\{E_n \leq b/n^p\}} \,\mathbb{1}_{\{n \leq T_{\mathcal{S}}\}} \,\mathbb{E}[Y_{n+1} \,|\, \mathcal{F}_n] \\
&\geq \mathbb{E}[Y_{n+1}^2 \,|\, \mathcal{F}_n] \\
&\quad\quad + 2\gamma_n E_n \,\mathbb{1}_{\{E_n \leq b/n^p\}} \,\mathbb{1}_{\{n \leq T_{\mathcal{S}}\}} \big[\beta E_n - 2\alpha(G^2 + \sigma^2)\gamma_n\big] \\
&\geq \mathbb{E}[Y_{n+1}^2 \,|\, \mathcal{F}_n] - 2\gamma_n \cdot b/n^p \cdot 2\alpha(G^2 + \sigma^2)\gamma_n \\
&= \mathbb{E}[Y_{n+1}^2 \,|\, \mathcal{F}_n] - 4\alpha^2 \beta^{-1}(G^2 + \sigma^2)^2 \gamma_n^3. \quad\quad \text{(C.31)}
\end{aligned}
$$

In view of the above, to establish (C.1), it suffices to show that $\mathbb{E}[Y_{n+1}^2 \,|\, \mathcal{F}_n] \geq B\gamma_n^2$ for some $B > 0$ and sufficiently large $n$. In this regard, Jensen's inequality gives

$$
\mathbb{E}[Y_{n+1}^2 \,|\, \mathcal{F}_n] \geq \mathbb{E}[Y_{n+1}^+ \,|\, \mathcal{F}_n]^2 \quad\quad \text{(C.32)}
$$

so it suffices to show that $\mathbb{E}[Y_{n+1}^+ \,|\, \mathcal{F}_n] = \Omega(\gamma_n)$. This is trivial if $n > T_{\mathcal{S}}$, so we are left to treat the case $n \leq T_{\mathcal{S}}$. For this case, (C.25) gives

$$
\mathbb{1}_{\{n \leq T_{\mathcal{S}}\}} \,\mathbb{E}[Y_{n+1}^+ \,|\, \mathcal{F}_n] \geq \mathbb{1}_{\{n \leq T_{\mathcal{S}}\}} \,\gamma_n \,\mathbb{E}[\psi_n^- \,|\, \mathcal{F}_n] - 2\,\mathbb{1}_{\{n \leq 2T_{\mathcal{S}}\}} \,\alpha(G^2 + \sigma^2)\gamma_n^2 \quad\quad \text{(C.33)}
$$

meaning that we need to focus on the expectation $\mathbb{E}[\psi_n^- \,|\, \mathcal{F}_n]$.

We consider two further cases (this is where Assumption 4 kicks in and plays a crucial role). First, if $X_n \notin \mathcal{M}$, Proposition C.1 and Assumption 5 applied to $v = -\nabla E(X_n)/\|\nabla E(X_n)\|$ give

$$
\begin{aligned}
\mathbb{1}_{\{n \leq T_{\mathcal{S}} \wedge X_n \notin \mathcal{M}\}} \,\mathbb{E}[\psi_n^- \,|\, \mathcal{F}_n] &= \mathbb{1}_{\{n \leq T_{\mathcal{S}} \wedge X_n \notin \mathcal{M}\}} \,\mathbb{E}[\langle -\nabla E(X_n), Z_n \rangle^+ \,|\, \mathcal{F}_n] \\
&\geq \mathbb{1}_{\{n \leq T_{\mathcal{S}} \wedge X_n \notin \mathcal{M}\}} \cdot c\|\nabla E(X_n)\| \\
&\geq \beta c \,\mathbb{1}_{\{n \leq T_{\mathcal{S}} \wedge X_n \notin \mathcal{M}\}}. \quad\quad \text{(C.34)}
\end{aligned}
$$

Otherwise, if $X_n \in \mathcal{M}$ (which, heuristically, should only happen with probability 0), choose a unit normal vector $u_n$ such that

$$
\langle u_n, z \rangle = 0 \quad \text{for all } z \in T_{X_n}\mathcal{M}. \quad\quad \text{(C.35)}
$$

Since the projector $P_{X_n}$ defined in (C.10) takes values in $T_{X_n}\mathcal{M}$, we will have $\langle u_n, P_{X_n}(Z_n) \rangle = 0$, and hence:

$$
\langle u_n, Z_n \rangle = \langle u_n, Z_n - P_{X_n}(Z_n) \rangle. \quad\quad \text{(C.36)}
$$

Therefore, by Proposition C.1, we get the chain of inequalities:

$$
\begin{aligned}
\mathbb{E}[[\nabla^+ E(X_n)[Z_n]]^- \,|\, \mathcal{F}_n] &\geq \beta \,\mathbb{E}[\|P_{X_n}(Z_n) - Z_n\| \,|\, \mathcal{F}_n] && \text{\{by Proposition C.1\}} \\
&\geq \beta \,\mathbb{E}[\langle u_n, Z_n - P_{X_n}(Z_n) \rangle^+ \,|\, \mathcal{F}_n] && \text{\{by Cauchy–Schwarz\}} \\
&= \beta \,\mathbb{E}[\langle u_n, Z_n \rangle^+ \,|\, \mathcal{F}_n] && \text{\{by (C.36)\}} \\
&\geq \beta c && \text{\{by Assumption 5\}}
\end{aligned}
$$

valid on the event $\{n \leq T_{\mathcal{S}} \wedge X_n \in \mathcal{M}\}$.

Putting together all of the above, we finally get

$$
\mathbb{1}_{\{n \leq T_{\mathcal{S}}\}} \,\mathbb{E}[\psi_n^- \,|\, \mathcal{F}_n] \geq \mathbb{1}_{\{n \leq T_{\mathcal{S}}\}} \,\beta c \quad\quad \text{(C.37)}
$$

and hence, by (C.33):

$$
\mathbb{E}[Y_{n+1}^+ \,|\, \mathcal{F}_n] \geq \beta c \gamma_n - 2\alpha(G^2 + \sigma^2)\gamma_n^2 = \Omega(\gamma_n) \qu\quad\quad \text{(C.38)}
$$

on the event $\{n \leq T_{\mathcal{S}}\}$. This completes our proof. ∎

# D  Rates of convergence

Our aim in this appendix is to establish the rate of convergence of (SGD) to local minima that are regular in the sense of Hurwicz, i.e., $H(x^*) \succ 0$. For convenience, we restate the relevant result below:

**Theorem 4.** *Fix some tolerance level $\delta > 0$, let $x^*$ be a regular minimizer of $f$, and suppose that Assumption 4 holds. Assume further that (SGD) is run with a step-size schedule of the form $\gamma_n = \gamma/(n+m)^p$ for some $p \in (2/(q+2), 1]$ and large enough $m, \gamma > 0$. Then:*

*1. There exist neighborhoods $\mathcal{U}$ and $\mathcal{U}_1$ of $x^*$ such that, if $X_1 \in \mathcal{U}_1$, the event*

$$E_\infty = \{X_n \in \mathcal{U} \text{ for all } n = 1, 2, \dots\} \tag{11}$$

*occurs with probability at least $1 - \delta$, i.e., $\mathbb{P}(E_\infty \mid X_1 \in \mathcal{U}_1) \geq 1 - \delta$.*

*2. Conditioned on $E$, we have*

$$\mathbb{E}[f(X_n) - f(x^*) \mid E_\infty] \leq \frac{2}{\beta} \frac{\gamma}{1-\delta} \frac{G^2 + \sigma^2}{2\alpha} \frac{1}{n^p} + o\left(\frac{1}{n^p}\right) \qquad \text{if } p < 1, \tag{12a}$$

*and*

$$\mathbb{E}[f(X_n) - f(x^*) \mid E_\infty] \leq \frac{2}{\beta} \frac{2\gamma^2}{1-\delta} \frac{G^2 + \sigma^2}{2\alpha\gamma - 1} \frac{1}{n} + o\left(\frac{1}{n}\right) \qquad \text{if } p = 1, \tag{12b}$$

*provided that $2\alpha\gamma > 1$ when $p = 1$.*

**Auxiliary results.**  The proof of Theorem 4 requires several ancillary results, which we state and prove below. The first is a lemma on numerical sequences, usually attributed to Chung [10]:

**Lemma D.1** (10, Lemma 1). *Let $a_n$, $n = 1, 2, \dots$, be a non-negative sequence such that*

$$a_{n+1} \leq \left[1 - \frac{P}{(n+m)^p}\right] a_n + \frac{R}{(n+m)^{p+r}} \tag{D.1}$$

*where $p \in (0, 1]$, $r > 0$ and $P, R > 0$. Then:*

*1. If $p < 1$, we have*

$$a_n \leq \frac{R}{P} \frac{1}{n^r} + o\left(\frac{1}{n^r}\right). \tag{D.2a}$$

*2. If instead $p = 1$ and $P > r$, we have*

$$a_n \leq \frac{R}{P - r} \frac{1}{n} + o\left(\frac{1}{n}\right). \tag{D.2b}$$

The next ingredient of the proof of Theorem 4 concerns the local behavior of $f$ near a regular minimizer; for convenience, we restate it below:

**Lemma 1.** *Let $x^*$ be a regular minimizer of $f$. Then, there exists a convex compact neighborhood $\mathcal{K}$ of $x^*$ and constants $\alpha, \beta > 0$ (possibly depending on $\mathcal{K}$) such that*

$$\alpha\|x - x^*\|^2 \leq f(x) - f(x^*) \leq \langle \nabla f(x), x - x^* \rangle \leq \beta\|x - x^*\|^2 \quad \text{for all } x \in \mathcal{K}. \tag{10}$$

*Proof.* Let $\mathcal{K}$ be a sufficiently small convex compact neighborhood of $x^*$ such that $H(x) \succ 0$ for all $x \in \mathcal{K}$ (that such a neighborhood exists is a consequence of the regularity of $x^*$ and the smoothness of $f$). Then, by compactness, there exist constants $\alpha, \beta$ such that $\alpha I \preccurlyeq H(x) \preccurlyeq \beta I$ for all $x \in \mathcal{K}$. Moreover, for all $x \in \mathcal{K}$, we have

$$\nabla f(x) = (x - x^*)^\top \int_0^1 H(x^* + t(x - x^*)) \, dt, \tag{D.3}$$

where we used the fact that $\nabla f(x^*) = 0$ (since $x^*$ is a minimizer of $f$). Hence, multiplying both sides by $x - x^*$, the mean value theorem for integrals yields:

$$\langle \nabla f(x), x - x^* \rangle = \int_0^1 (x - x^*)^\top H(x^* + t(x - x^*))(x - x^*) \, dt$$

$$= (x - x^*)^\top H(x')(x - x^*) \tag{D.4}$$

for some $x' \in [x^*, x]$. Since $\alpha I \preccurlyeq H(x') \preccurlyeq \beta I$, our claim follows. ∎

Thanks to Lemma 1, we obtain the following recursive estimate for (SGD):

**Proposition D.1.** *Let $x^*$ be a regular minimum of $f$ and let $\mathcal{K}$ and $\alpha$ be as in Lemma 1. Assume moreover that $X_n \in \mathcal{K}$ for some $n \geq 1$ and let*

$$D_n = \frac{1}{2}\|X_n - x^*\|^2. \tag{D.5}$$

*We then have:*

$$D_{n+1} \leq (1 - 2\alpha\gamma_n)D_n + \gamma_n\xi_n + \frac{1}{2}\gamma_n^2\|V_n\|^2, \tag{D.6}$$

*where $\xi_n = -\langle Z_n, X_n - x^*\rangle$ is a martingale difference sequence.*

*Proof.* Recall first that $X_{n+1} = X_n - \gamma_n(\nabla f(X_n) + Z_n)$ where $Z_n$ is the gradient error at $X_n$. Then, by the definition of $D_n$, we have:

$$
\begin{aligned}
D_{n+1} = \frac{1}{2}\|X_{n+1} - x^*\|^2 &= \frac{1}{2}\|X_n - x^* - \gamma_n V_n\|^2 \\
&= \frac{1}{2}\|X_n - x^*\|^2 - \gamma_n\langle V_n, X_n - x^*\rangle + \frac{1}{2}\gamma_n^2\|V_n\|^2 \\
&= D_n - \gamma_n\langle\nabla f(X_n), X_n - x^*\rangle - \gamma_n\langle Z_n, X_n - x^*\rangle + \frac{1}{2}\gamma_n^2\|V_n\|^2 \\
&\leq D_n - \alpha\gamma_n\|X_n - x^*\|^2 + \gamma_n\xi_n + \frac{1}{2}\gamma_n^2\|V_n\|^2 \\
&= (1 - 2\alpha\gamma_n)D_n + \gamma_n\xi_n + \frac{1}{2}\gamma_n^2\|V_n\|^2
\end{aligned}
\tag{D.7}
$$

where the second-to-last line follows from Lemma 1. Since $\mathbb{E}[\xi_n \,|\, \mathcal{F}_n] = \langle \mathbb{E}[Z_n \,|\, \mathcal{F}_n], X_n - x^*\rangle = 0$, our claim follows (recall here that, by definition, $Z_n$ is not $\mathcal{F}_n$-measurable but $X_n$ is). ∎

With these basic results at our disposal, the proof of Theorem 4 will roughly follow the technical trajectory outlined below:

1. By Proposition D.1, $D_n$ grows at most by $\gamma_n\xi_n + \frac{1}{2}\gamma_n^2\|V_n\|^2$ at each step. This quantity can be big for any given $n$ but we will show that, with high probability (and, in particular, with probability at least $1 - \delta$), the aggregation of these errors remains controllably small. This will be the most technical and involved part of our argument.

2. Using the above, we will show that, with probability at least $1 - \delta$, $D_n$ cannot grow more than a token quantity $\varepsilon$. As a result, if the initial distance to $x^*$ is not too big, $X_n$ will remain in a neighborhood thereof for all time.

3. For the final part of the theorem, we will condition on this event to map (D.6) to a recursion of the form (D.1), and we will subsequently employ Lemma D.1 to obtain the stated result. The main problem here is that, after conditioning, the noise in (D.6) is no longer zero-mean, so we will need to adapt our analysis to the new noise distribution.

We make all this precise below. For convenience, we focus on the case $p > 1/2$; the case $p \in (2/(q+2), 1/2]$ follows by modifying the arguments that follow with the Hölder estimates we introduced in the proof of Lemma B.1.

**Controlling the error terms.** We begin by encoding the error terms in (D.6) as

$$M_n = \sum_{k=1}^{n} \gamma_k\xi_k \tag{D.8}$$

and

$$S_n = \frac{1}{2}\sum_{k=1}^{n} \gamma_k^2\|V_k\|^2 \tag{D.9}$$

Since $\mathbb{E}[\xi_n \,|\, \mathcal{F}_n] = 0$, we have $\mathbb{E}[M_n \,|\, \mathcal{F}_n] = M_{n-1}$, so $M_n$ is a zero-mean martingale; likewise, $\mathbb{E}[S_n \,|\, \mathcal{F}_n] \geq S_{n-1}$, so $S_n$ is a submartingale. Interestingly, even though $M_n$ is more "neutral" as an error (because $\xi_n$ is zero-mean), it is more difficult to control because the variance of its increments is

$$\mathbb{E}[\|\gamma_n\xi_n\|^2 \,|\, \mathcal{F}_n] = \gamma_n^2\,\mathbb{E}[\langle Z_n, X_n - x^*\rangle^2 \,|\, \mathcal{F}_n] \tag{D.10}$$

and this last quantity can become arbitrarily big if $X_n$ does not remain in the vicinity of $x^*$ (which is what we are trying to prove). Because of this, we need to take a less direct, step-by-step approach to bound the total error increments *conditioned* on the event that $X_n$ remains close to $x^*$.

To formalize this, introduce the "cumulative mean square" error

$$R_n = M_n^2 + S_n. \tag{D.11}$$

By construction, we have

$$\begin{aligned} R_n &= (M_{n-1} + \gamma_n \xi_n)^2 + S_{n-1} + \tfrac{1}{2}\gamma_n^2 \|V_n\|^2 \\ &= R_{n-1} + 2M_{n-1}\gamma_n \xi_n + \gamma_n^2 \xi_n^2 + \tfrac{1}{2}\gamma_n^2 \|V_n\|^2 \end{aligned} \tag{D.12}$$

and hence, after taking expectations:

$$\mathbb{E}[R_n \mid \mathcal{F}_n] = R_{n-1} + 2M_{n-1}\gamma_n \,\mathbb{E}[\xi_n \mid \mathcal{F}_n] + \gamma_n^2 \,\mathbb{E}[\xi_n^2 + \tfrac{1}{2}\|V_n\|^2 \mid \mathcal{F}_n] \geq R_{n-1} \tag{D.13}$$

i.e., $R_n$ is a submartingale. To condition it further, let $\mathcal{U}$ be a neighborhood of $x^*$, let $\varepsilon > 0$, and define the events

$$E_n \equiv E_n(\mathcal{U}) = \{X_n \in \mathcal{U} \text{ for all } k = 1, 2, \ldots, n\} \tag{D.14}$$

and

$$H_n \equiv H_n(\varepsilon) = \{R_k \leq \varepsilon \text{ for all } k = 1, 2, \ldots, n\}. \tag{D.15}$$

By definition, we also have $E_0 = H_0 = \Omega$ (because the set-building index set for $k$ is empty in this case, and every statement is true for the elements of the empty set). These events will play a crucial role in the sequel as indicators of whether $X_n$ has escaped the vicinity of $x^*$ or not.

To proceed, we will instantiate $\mathcal{U}$ and $\varepsilon$ in the definition of $E$ and $H$ respectively as follows. First, for (D.14), we will choose a neighborhood $\mathcal{U}$ contained in the convex compact neighborhood $\mathcal{K}$ of $x^*$ (whose existence is guaranteed by Lemma 1); in particular, this implies that (10) holds for all $x \in \mathcal{U}$. Moreover, with a fair degree of hindsight, we will also choose $\varepsilon > 0$ such that

$$\{x \in \mathbb{R}^d : \|x - x^*\|^2 \leq 4\varepsilon + 2\sqrt{\varepsilon}\} \subseteq \mathcal{U}. \tag{D.16}$$

and we will assume that $X_1$ is initialized in a neighborhood $\mathcal{U}_1 \subseteq \mathcal{U}$ such that

$$\mathcal{U}_1 \subseteq \{x \in \mathbb{R}^d : \|x - x^*\|^2 \leq 2\varepsilon\} \tag{D.17}$$

These will be the neighborhoods $\mathcal{U}$ and $\mathcal{U}_1$ whose existence is postulated by Theorem 4. Then, with all this in hand, we have:

**Lemma D.2.** *Let $x^*$ be a regular minimizer of $f$ as above and assume that Assumption 4 holds. Then, for all $n = 1, 2, \ldots$, we have:*

1. *$E_{n+1} \subseteq E_n$ and $H_{n+1} \subseteq H_n$.*

2. *$H_{n-1} \subseteq E_n$.*

3. *Consider the "large noise" event*

$$\tilde{H}_n \equiv H_{n-1} \backslash H_n = H_{n-1} \cap \{R_n > \varepsilon\} = \{R_k \leq \varepsilon \text{ for all } k = 1, 2, \ldots, n-1 \text{ and } R_n > \varepsilon\}, \tag{D.18}$$

   *and let $\tilde{R}_n = R_n \mathbb{1}_{H_{n-1}}$ denote the cumulative error subject to the noise being "small" until time $n$. Then:*

$$\mathbb{E}[\tilde{R}_n] \leq \mathbb{E}[\tilde{R}_{n-1}] + [G^2 + (1 + r_{\mathcal{U}}^2)\sigma^2]\gamma_n^2 - \varepsilon \mathbb{P}(\tilde{H}_{n-1}), \tag{D.19}$$

   *where $r_{\mathcal{U}} = \sup_{x \in \mathcal{U}} \|x - x^*\|$ and, by convention, we write $\tilde{H}_0 = \varnothing$ and $\tilde{R}_0 = 0$.*

*Remark.* In the above (and what follows), the notation $\mathbb{1}_A$ is used to indicate the logical indicator of an event $A \subseteq \Omega$, i.e., $\mathbb{1}_A(\omega) = 1$ if $\omega \in A$ and $\mathbb{1}_A(\omega) = 0$ otherwise.

*Proof.* The first claim is obvious. For the second, we proceed inductively:

1. For the base case $n = 1$, we have $E_1 = \{X_1 \in \mathcal{U}\} \supseteq \{X_1 \in \mathcal{U}_1\} = \Omega$ because $X_1$ is initialized in $\mathcal{U}_1 \subseteq \mathcal{U}$. Since $H_0 = \Omega$, our claim follows.

2. For the inductive step, assume that $H_{n-1} \subseteq E_n$ for some $n \geq 1$. To show that $H_n \subseteq E_{n+1}$, fix a realization in $H_n$ so $R_k \leq \varepsilon$ for all $k = 1, 2, \ldots, n$. Since $H_n \subseteq H_{n-1}$, the inductive hypothesis posits that $E_n$ also occurs, i.e., $X_k \in \mathcal{U}$ for all $k = 1, 2, \ldots, n$; hence, it suffices to show that $X_{n+1} \in \mathcal{U}$.

To that end, given that $X_k \in \mathcal{U} \subseteq \mathcal{K}$ for all $k = 1, 2, \ldots n$, the distance estimate (D.6) readily gives

$$D_{k+1} \leq D_k + \gamma_k \xi_k + \tfrac{1}{2}\gamma_k^2 \|V_k\|^2 \quad \text{for all } k = 1, 2, \ldots n. \tag{D.20}$$

Therefore, after telescoping, we obtain

$$D_{n+1} \leq D_1 + M_n + S_n \leq D_1 + \sqrt{R_n} + R_n \leq \varepsilon + \sqrt{\varepsilon} + \varepsilon = 2\varepsilon + \sqrt{\varepsilon} \tag{D.21}$$

by the inductive hypothesis. We conclude that $\|X_{n+1} - x^*\|^2 = 2D_{n+1} \leq 4\varepsilon + 2\sqrt{\varepsilon}$, so $X_{n+1} \in \mathcal{U}$ and the induction is complete.

For our third claim, we decompose $\tilde{R}_n$ as

$$
\begin{aligned}
\tilde{R}_n = R_n \mathbb{1}_{H_{n-1}} = R_{n-1} \mathbb{1}_{H_{n-1}} &+ (R_n - R_{n-1}) \mathbb{1}_{H_{n-1}} \\
&= R_{n-1} \mathbb{1}_{H_{n-2}} - R_{n-1} \mathbb{1}_{\tilde{H}_{n-1}} + (R_n - R_{n-1}) \mathbb{1}_{H_{n-1}}, \\
&= \tilde{R}_{n-1} + (R_n - R_{n-1}) \mathbb{1}_{H_{n-1}} - R_{n-1} \mathbb{1}_{\tilde{H}_{n-1}},
\end{aligned}
\tag{D.22}
$$

where we used the fact that $H_{n-1} = H_{n-2} \setminus \tilde{H}_{n-1}$ so $\mathbb{1}_{H_{n-1}} = \mathbb{1}_{H_{n-2}} - \mathbb{1}_{\tilde{H}_{n-1}}$ (recall here that $H_{n-1} \subseteq H_{n-2}$). Now, to proceed, (D.12) yields

$$R_n - R_{n-1} = 2M_{n-1}\gamma_n \xi_n + \gamma_n^2 \xi_n^2 + \tfrac{1}{2}\gamma_n^2 \|V_n\|^2 \tag{D.23}$$

so

$$\mathbb{E}[(R_n - R_{n-1}) \mathbb{1}_{H_{n-1}}] = 2\gamma_n \mathbb{E}[M_{n-1}\xi_n \mathbb{1}_{H_{n-1}}] \tag{D.24a}$$
$$+ \gamma_n^2 \mathbb{E}[\xi_n^2 \mathbb{1}_{H_{n-1}}] \tag{D.24b}$$
$$+ \tfrac{1}{2}\gamma_n^2 \mathbb{E}[\|V_n\|^2 \mathbb{1}_{H_{n-1}}] \tag{D.24c}$$

However, since $H_{n-1}$ and $M_{n-1}$ are both $\mathcal{F}_n$-measurable, we have the following estimates:

1. For the noise term in (D.24a), the second part of Proposition D.1 gives:

$$\mathbb{E}[M_{n-1}\xi_n \mathbb{1}_{H_{n-1}}] = \mathbb{E}[M_{n-1} \mathbb{1}_{H_{n-1}} \mathbb{E}[\xi_n \mid \mathcal{F}_n]] = 0. \tag{D.25}$$

2. The term (D.24b) is where the conditioning on $H_{n-1}$ plays the most important role because it allows us to control the distance $\|X_n - x^*\|$. Specifically, we have:

$$
\begin{aligned}
\mathbb{E}[\xi_n^2 \mathbb{1}_{H_{n-1}}] &= \mathbb{E}[\mathbb{1}_{H_{n-1}} \mathbb{E}[\langle Z_n, X_n - x^* \rangle^2 \mid \mathcal{F}_n]] \\
&\leq \mathbb{E}[\mathbb{1}_{H_{n-1}} \|X_n - x^*\|^2 \mathbb{E}[\|Z_n\|^2 \mid \mathcal{F}_n]] &&\{\text{by Cauchy–Schwarz}\} \\
&\leq \mathbb{E}[\mathbb{1}_{E_n} \|X_n - x^*\|^2 \mathbb{E}[\|Z_n\|^2 \mid \mathcal{F}_n]] &&\{\text{because } H_{n-1} \subseteq E_n\} \\
&\leq r_{\mathcal{U}}^2 \sigma^2. &&\{\text{by Assumption 4}\}
\end{aligned}
$$

3. Finally, for the term (D.24c), we have:

$$\mathbb{E}[\|V_n\|^2 \mathbb{1}_{H_{n-1}}] \leq \mathbb{E}[\|V_n\|^2] \leq 2\mathbb{E}[\|\nabla f(X_n)\|^2 + \|Z_n\|^2] \leq 2(G^2 + \sigma^2), \tag{D.26}$$

with the last step following from Assumptions 1 and 4.

Thus, putting together all of the above, we obtain:

$$\mathbb{E}[(R_n - R_{n-1}) \mathbb{1}_{H_{n-1}}] \leq [G^2 + (1 + r_{\mathcal{U}}^2)\sigma^2]\gamma_n^2 \tag{D.27}$$

Going back to (D.22), we have $R_{n-1} > \varepsilon$ if $\tilde{H}_{n-1}$ occurs, so the last term becomes

$$\mathbb{E}[R_{n-1} \mathbb{1}_{\tilde{H}_{n-1}}] \geq \varepsilon \mathbb{E}[\mathbb{1}_{\tilde{H}_{n-1}}] = \varepsilon \mathbb{P}(\tilde{H}_{n-1}). \tag{D.28}$$

Our claim then follows by combining Eqs. (D.22), (D.26) and (D.28). ∎

**Controlling the probability of escape.** Lemma D.2 is the technical key to show that $X_n$ remains close to $x^*$ with high probability; we formalize this in a final intermediate result below.

**Proposition D.2.** *Fix some tolerance level $\delta > 0$. If Assumption 4 holds and (SGD) is run with a step-size schedule of the form $\gamma_n = \gamma/(n+m)^p$ for some sufficiently large $m > 0$, we have*

$$\mathbb{P}(H_n) \geq 1 - \delta \quad \text{for all } n = 1, 2, \dots \tag{D.29}$$

*Proof.* We begin by bounding the probability of the "large noise" event $\tilde{H}_n = H_{n-1} \setminus H_n$ as follows:

$$
\begin{aligned}
\mathbb{P}(\tilde{H}_n) = \mathbb{P}(H_{n-1} \setminus H_n) &= \mathbb{P}(H_{n-1} \cap \{R_n > \varepsilon\}) \\
&= \mathbb{E}[\mathbb{1}_{H_{n-1}} \times \mathbb{1}_{\{R_n > \varepsilon\}}] \\
&\leq \mathbb{E}[\mathbb{1}_{H_{n-1}} \times (R_n/\varepsilon)] \\
&= \mathbb{E}[\tilde{R}_n]/\varepsilon
\end{aligned}
\tag{D.30}
$$

where, in the second-to-last line, we used the fact that $R_n \geq 0$ (so $\mathbb{1}_{\{R_n > \varepsilon\}} \leq R_n/\varepsilon$). Now, by telescoping (D.19), we get

$$\mathbb{E}[\tilde{R}_n] \leq \mathbb{E}[\tilde{R}_0] + R_* \sum_{k=1}^{n} \gamma_k^2 - \varepsilon \sum_{k=1}^{n} \mathbb{P}(\tilde{H}_{k-1}) \tag{D.31}$$

where we set $R_* = G^2 + (1 + r_{\mathcal{U}}^2)\sigma^2$. Hence, combining (D.30) and (D.31), we obtain the estimate

$$\sum_{k=1}^{n} \mathbb{P}(\tilde{H}_k) \leq \frac{R_*}{\varepsilon} \sum_{k=1}^{n} \gamma_k^2 \leq \frac{R_* \Gamma}{\varepsilon}, \tag{D.32}$$

where we set $\Gamma = \sum_{n=1}^{\infty} \gamma_n^2 = \gamma^2 \sum_{n=1}^{\infty} (n+m)^{-2p}$ and we used the fact that $\tilde{R}_0 = 0$ and $\tilde{H}_0 = \varnothing$ (by convention).

By choosing $m$ sufficiently large, we can ensure that $R_* \Gamma/\varepsilon < \delta$; moreover, since the events $\tilde{H}_k$ are disjoint for all $k = 1, 2, \dots$, we get

$$\mathbb{P}\left( \bigcup_{k=1}^{n} \tilde{H}_k \right) = \sum_{k=1}^{n} \mathbb{P}(\tilde{H}_k) \leq \delta \tag{D.33}$$

and hence:

$$\mathbb{P}(H_n) = \mathbb{P}\left( \bigcap_{k=1}^{n} \tilde{H}_k^{\mathrm{c}} \right) \geq 1 - \delta, \tag{D.34}$$

as claimed. ∎

**Putting everything together.** We are finally in a position to combine all of the ingredients for the proof of Theorem 4.

*Proof of Theorem 4.* To begin, define $\mathcal{U}$ and $\mathcal{U}_1$ as in Lemma D.2. Then, by construction, we have:

$$E_\infty \equiv \{X_n \in \mathcal{U} \text{ for all } n = 1, 2, \dots\} = \bigcap_{n=1}^{\infty} E_n. \tag{D.35}$$

Since the sequence $E_n$ is decreasing and $E_n \supseteq H_{n-1}$ (by the second part of Lemma D.2), Proposition D.2 yields

$$\mathbb{P}(E_\infty) = \inf_n \mathbb{P}(E_n) \geq \inf_n \mathbb{P}(H_{n-1}) \geq 1 - \delta, \tag{D.36}$$

provided that $m$ is chosen large enough. This proves the first part of the theorem, i.e., to the effect that $X_n$ remains close to $x^*$ with probability at least $1 - \delta$.

For the second part of the theorem, Proposition D.1 readily gives

$$D_{n+1} \mathbb{1}_{E_n} \leq (1 - 2\alpha\gamma_n) D_n \mathbb{1}_{E_n} + [\gamma_n \xi_n + \tfrac{1}{2}\gamma_n^2 \|V_n\|^2] \mathbb{1}_{E_n}. \tag{D.37}$$

Now, for any given $\gamma$, we can choose $m$ sufficiently large so that $\inf_n(1 - 2\alpha\gamma_n) > 0$ and $\mathbb{P}(E_\infty) \geq 1 - \delta$ (the latter by Proposition D.2). Moreover, working as in the proof of Lemma D.2, we get

$$\mathbb{E}\big[(\gamma_n\xi_n + \tfrac{1}{2}\|V_n\|^2)\,\mathbb{1}_{E_n}\big] = \mathbb{E}\big[\mathbb{1}_{E_n}\,\mathbb{E}\big[\gamma_n\xi_n + \tfrac{1}{2}\gamma_n^2\|V_n\|^2\,\big|\,\mathcal{F}_n\big]\big]$$
$$\leq \gamma_n^2(G^2 + \sigma^2). \tag{D.38}$$

Then, letting $\bar{D}_n = \mathbb{E}[D_n\,\mathbb{1}_{E_n}] \geq 0$ and recalling that $E_{n+1} \subseteq E_n$ (so $\mathbb{1}_{E_{n+1}} \leq \mathbb{1}_{E_n}$), the two estimates above yield

$$\bar{D}_{n+1} \leq \mathbb{E}[D_{n+1}\,\mathbb{1}_{E_n}] \leq (1 - 2\alpha\gamma_n)\bar{D}_n + (G^2 + \sigma^2)\gamma_n^2$$
$$\leq \left[1 - \frac{2\alpha\gamma}{(n+m)^p}\right]\bar{D}_n + \frac{(G^2 + \sigma^2)\gamma^2}{(n+m)^{2p}}. \tag{D.39}$$

Thus, by Lemma D.1, we obtain the bounds:

$$\bar{D}_n \leq \frac{G^2 + \sigma^2}{2\alpha}\frac{\gamma}{n^p} + o\left(\frac{1}{n^p}\right) \qquad \text{if } p < 1, \tag{D.40a}$$

and

$$\bar{D}_n \leq \frac{G^2 + \sigma^2}{2\alpha\gamma - 1}\frac{\gamma^2}{n} + o\left(\frac{1}{n^p}\right) \qquad \text{if } p = 1, \tag{D.40b}$$

provided that $2\alpha\gamma > 1$ for the latter. The claim of the theorem then follows by noting that

$$\mathbb{E}[\|X_n - x^*\|^2\,|\,E_\infty] \leq \frac{\mathbb{E}[\|X_n - x^*\|^2\,\mathbb{1}_{E_\infty}]}{\mathbb{P}(E_\infty)} \leq \frac{2}{1-\delta}\bar{D}_n \tag{D.41}$$

and applying Eqs. (D.40a) and (D.40b). ∎

# E   Numerical experiments

In this appendix, we present some more details on our ResNet training setup and some additional numerical results. We used the *python/pytorch* implementation of Resnet18 from the *torchvision* package and, for consistency, we downloaded the CIFAR10 dataset from the same package. For training/evaluation purposes, we used the the standard training/test split of 50000/10000 examples, with training and test batches of size 120.

The purposes of our experiments is to demonstrate the possible benefits of the "cooldown" heuristic that is derived from our convergence analysis in Section 4. To that end, we initially trained the model with constant step-size (SGD) whose step-size is picked through grid-search over the set $\{1, 10^{-1}, 10^{-2}, 10^{-3}, 10^{-4}\}$. We then took checkpoints of the model at certain epochs and launched the cooldown heuristic from such points with a $1/n$ step-size policy. It is important to emphasize that the iteration counter $n$ starts at the first iteration of cooldown phase so that we have a "continuous" sequence of step-sizes across epochs. In Fig. 3, we provide the complementary plots for the setting described in Section 5, again exhibiting a clear benefit (especially in test accuracy and loss) when using the cool-down heuristic for the last part of the experiment's runtime budget.

**(a)** Training loss

**(b)** Training accuracy

**(c)** Test loss

**(d)** Test accuracy

**Figure 3:** Results for training ResNet18 model for classification over CIFAR10 dataset, with cooldown heuristic. Constant step-size SGD is run for 100 epochs and cooldown phase starts at epochs 70, 80, 90 and 95, with diminishing step-size policy of $1/n$.