[Reviews · NeurIPS 2020]

Review 1

Summary and Contributions: This paper proves that with probability 1, SGD will converge to a point that is not strict saddle points. It also provides a local non-asymptotic convergence if initial point is located in a local neighborhood of a strongly convex local minimizer.

Strengths: The probability 1 result is technically new.

Weaknesses: There are a lot similar results in slightly different regime, which makes this work looks incremental. (a) [17, 18] already showed that with probability, GD will converge to a point that is not strict saddle point, the only difference of this work is that this work adds the noise in gradients. (b) [27] already proves morally the same result only under a bit restrictive Morse assumption. In the case of GD, this Morse assumption can be resolved by using a stronger stable manifold theorem in "Michael Shub. Global stability of dynamical systems.", I suspect a similar combination might go through here? (c) there are already several non-asymptotic results with sharp rate for SGD in avoiding saddle point [14 and two additional works in "Relation to prior work" section]. Usually one view asymptotic results (this paper) weaker than non-asymptotic results (earlier papers), it is also not clear from this paper if one can obtain probability 1 result by modifying the existing high probability result with Borel Cantelli lemma and a bit extra work.

Correctness: Yes

Clarity: Yes

Relation to Prior Work: There are two recent SGD analyses that are relevant to this paper: "On Nonconvex Optimization for Machine Learning: Gradients, Stochasticity, and Saddle Points"; "Sharp analysis for nonconvex SGD escaping from saddle points".

Reproducibility: Yes

Additional Feedback:


Review 2

Summary and Contributions: This paper analyzes the orbit of stochastic gradient descent and the main result is that SGD avoids saddle point and the interpolated continuous trajectory converges almost surely. The authors give a detailed literature review and comparison to previous work. The technique used in this paper is the combination of classic results from dynamical system and probability theory. This approach is rare in machine learning literature. Even considering the amount of research in this area (convergence and saddle avoidance for GD), the result of this paper is of some interest.

Strengths: Saddle point avoidance results are important in non-convex optimizations and significant amount of research(asymptotic avoidance and non-asymptotic rate of escaping) is focusing on this area, the results is relevant to machine learning community.

Weaknesses: 1. The selling point of this paper. The authors emphasize on the convergence of SGD which is presented in theorem 2 and corollary 2. But I think this is somehow captured by the literature like “Stochastic gradient on Riemannian manifold, Bonnabel 2013”, where the almost surely convergence is established even for Riemannian manifold? Convergence of SGD in Euclidean space might be derived from this paper. From this perspective, convergence result lacks the novelty as it is stated. 2. The novelty of techniques. It seems that the techniques and results of this paper are very incremental and almost the same with that of Pemantle's. 3. The rate of convergence. In theorem 4, I cannot see a practical way to choose parameters to have a convergence guarantee as Jin et al., so this is still an asymptotic result.

Correctness: Correct

Clarity: The paper is well written and easy to follow.

Relation to Prior Work: The authors have reviewed most well known works in the literature.

Reproducibility: Yes

Additional Feedback: Question to be addressed: Please compare the assumptions listed in this paper to the assumptions listed by Pemantle [27] (iii and iv, theorem 1) to give a clearer view for the avoidance results.


Review 3

Summary and Contributions: This paper considers the convergence of stochastic gradient descent (SGD) for nonconvex problems. The main contribution includes proving that (1) SGD converges to a critical point (subsequence convergence) almost surely (2) SGD also escapes strict saddles almost surely, (3) local convergence to a strict minimizer once the iterates get close to it.

Strengths: 1. It is a theoretical paper, providing new convergence results for SGD when used for solving nonconvex problem. 2. Escaping strict saddle points of SGD without the help of additional noise seems novel and can provide theoretical guarantee for practical use of SGD. 3. The convergence rate to local minimum is provided, but suffers few weaknesses which are described below.

Weaknesses: 1. Compared to Ge et al. [10] which provides the rate of escaping saddle points of SGD, there is no explicit rate of escaping saddle points provided in this work. 2. Theorem 4 provides the rate of convergence to local minimum, but (1) it only works for strict local minimizers whose Hessian are positive definite, (2) it only provides the rate once the iterate is close to the strict local minimizer, but there is no guarantee if this happens, or how fast it happens. 3. The following work is very related and is missing: Daneshmand, Hadi, Jonas Kohler, Aurelien Lucchi, and Thomas Hofmann. "Escaping Saddles with Stochastic Gradients." In International Conference on Machine Learning, pp. 1155-1164. 2018.

Correctness: They are correct, but I didn't carefully check the proofs.

Clarity: The paper is well written

Relation to Prior Work: The following work is very related and is missing: Daneshmand, Hadi, Jonas Kohler, Aurelien Lucchi, and Thomas Hofmann. "Escaping Saddles with Stochastic Gradients." In International Conference on Machine Learning, pp. 1155-1164. 2018.

Reproducibility: Yes

Additional Feedback: _______________________________________After rebuttal_______________________ I appreciate the effort the authors have made to address my comment. In Theorem 4, the dependence of m (or other parameters) on $\delta$ should be highlighted since in the current version, one can simply set $\delta = 0$, which gives probability of 1.


Review 4

Summary and Contributions: This paper shows that under mild conditions, SGD converges to a critical point of the general non-convex functions and avoids all strict saddle points, with probability 1. It also presents a convergence analysis of the SGD once it enters the neighborhood of a local minimum.

Strengths: The paper generalizes the known convergence results for the deterministic GD and the SGD. The theoretical analysis is novel and elegant. It combines a series of results in the theory of dynamical systems and shows that they can bridge the continuous GD trajectory to the SGD trajectory. I like the APT analysis. By showing a relative weak property that the SGD is always bounded, it automatically gives the strong conclusion that SGD converges to the same set as the continuous GD.

Weaknesses: - The rate of convergence analysis would be more interesting if it bounds the iteration complexity before hitting the neighborhood U1. - What's the convergence rate's dependence on the dimension d?

Correctness: To the best of my knowledge, they are correct.

Clarity: The paper is very well written.

Relation to Prior Work: Yes

Reproducibility: Yes

Additional Feedback: Overall, this is an interesting paper. The paper presents a set of useful tools and may provide new insights to the study of non-convex optimization theory.

[Author Response · NeurIPS 2020]

We are grateful to the reviewers for their time and comments. For the reviewers' convenience, we briefly state below
the novel contributions of our work, as summarized by R4 (whom we thank for the expert summary):

> 3 "*This paper shows that under mild conditions, SGD converges to a critical point of general non-*
> 4 *convex functions and avoids all strict saddle points, with probability* 1. *It also presents a convergence*
> 5 *rate analysis of SGD once it enters the neighborhood of a local minimum.*"

In what follows, we address the reviewers' comments in order, tagging the reviewers concerned in each as **#RX**.

**#R1: Relation to Lee et al [17,18].** As we explain in Lines 66-68 of the introduction, **[17,18] do not study stochastic**
**gradient descent,** but gradient descent with full, *perfect* gradients – i.e., a ***deterministic*** algorithm. Specifically, [17,18]
show that **deterministic** gradient descent avoids strict saddles from almost every initial condition. The reviewer is
therefore not correct in interpreting this statement as an "in probability" result for SGD: *the results of [17,18] provide*
***no*** *guarantees for SGD, from* **any** *initial condition.*

*Additional comments:* The stochasticity in SGD makes for a drastically different, much more difficult setting. In the full
gradient case, there is a well-defined drift that drives GD away from saddle points. This persistent push is no longer
present in SGD: this is a crucial difference which we feel may be at the source of this misunderstanding.

**#R1#R2: Relation to Pemantle [27].** Pemantle's work only applies to **isolated, linearly unstable saddle points,** it
does not cover saddle points with a **non-trivial center manifold.** In the deterministic case, strict saddles can indeed
be excluded thanks to the existence of local diffeomorphism results based on the center manifold theorem. However,
in the stochastic case, the presence of a non-trivial center manifold requires fundamentally different techniques from
differential geometry, as we explain in detail in Appendices D.2 and D.3. The reason for this is that there is no longer
a persistent drift away from the center stable manifold (in technical terms, there is no "shadowing"). **This major**
**difficulty is not present in Pemantle's work** (which, again, cannot deal with non-trivial center manifolds); the only
relation with [27] is two technical lemmas on random numerical sequences (Lemmas D.1 and D.2).

*Additional comments:* The reviewers may have thought that we are making a significantly more restrictive Morse-Smale
assumption for the problem's objective – we emphasize here that *this is not the case.*

**#R1#R2: On the rates of Jin et al [14].** First, as can be seen from (E.3) and (E.42), Thm. 4 gives the precise bound

$$\mathbb{E}[f(X_n) - f(x^*) \mid X_1 \in \mathcal{U}_1] \le \frac{2}{\beta} \frac{2\gamma^2}{1-\delta} \frac{G^2+\sigma^2}{2\alpha\gamma-1} \frac{1}{n} + o\left(\frac{1}{n}\right). \tag{1}$$

We will put this expression for $p = 1$ in the main text. Beyond this, there are two key factual misunderstandings:

1. **The statements for SGD in [14] and related papers are also asymptotic** because they involve an unknown,
   probabilistic constant hidden in the $\mathcal{O}(\cdot)$ notation; see Theorem 3, Corollary 4 and Theorem 5 in [14], as well as
   the corresponding statements in the papers mentioned by R1.
2. The asymptotic value convergence rate of [14] and related papers is $\mathcal{O}(1/\sqrt{n})$; by contrast, **the value convergence**
   **guarantee that we provide is $\mathcal{O}(1/n)$.** The reviewers are therefore incorrect in stating that our rates are similar
   to those of [14] and related works.

33 **#R1: From high probability to probability 1 via Borel-Cantelli.** This is not possible for (at least) two reasons:

1. The target probability threshold $\zeta$ of Ge et al. is hard-coded in the algorithm's step-size. Therefore, getting
   results for different probability thresholds (in order to apply Borel-Cantelli) would necessitate running different
   algorithms, destroying in this way the validity of the results of Ge et al.
2. Even if this vital obstacle were to be somehow overcome, the logarithmic dependence of the step-size of Ge et
   al. on $\zeta$ implies that the induced step-size policy would have to vanish at an exponential rate in order to apply
   Borel-Cantelli. However, it is well known from standard results in stochastic approximation that SGD with
   summable step-size policies *does not converge* (Kushner and Yin, 1997, Chap. 4), so this approach would fail.

41 **#R2: On Bonnabel (2013).** We thank R2 for bringing this paper to our attention, we will definitely discuss it! At
42 the same time, we should point out that **Bonnabel's paper makes the explicit assumption that SGD remains in a**
43 **compact set** (cf. Theorems 1 and 2). Boundedness assumptions of this kind are prevalent in the literature, and **this is**
44 **precisely one of the key gaps that our paper closes:** convergence of SGD *without* implicit, unverifiable boundedness
45 assumptions. This was the main weakness identified by R2, so we hope that the above clarifies the merits of our work.

46 **#R3: On the rates of escape.** Deriving rates of escape that hold with probability 1 is a whole new paper in itself.

47 **#R3: On the size of $\mathcal{U}_1$.** The size of $\mathcal{U}_1$ only depends on the landscape of $f$ around $x^*$, *not* $\delta$; see (E.17) and (E.18).

48 **#R3: On Dashamand et al.** Dashamand et al. refine the analysis of Ge et al. and provide positive probability results
49 for second-order stationary points. There is no overlap with our techniques or results; we will cite it to make this clear.

50 **#R4: On the hitting time to $\mathcal{U}_1$.** This is a very difficult global-to-local estimate. To the best of our knowledge, no one
51 has succeeded in making progress on similar questions in general non-convex settings, so we do not address this here.

52 **#R4: On the dependence on $d$.** Great question! The rate does not explicitly depend on $d$, see (1) above.

[Meta-Review · NeurIPS 2020]

The paper has claimed three contributions, answering three questions. 1) Convergence to critical points without “bounded iterates” assumptions. In general, I think removing boundedness assumption is useful. But Reviewer 3 pointed out that this work adds the "bounded level set" assumption and bounded gradient assumption, which do not seem much stronger than “bounded iterates”. In particular, the gap between “bounded level-set” and “bounded iterates” is quite technical (not necessarily small, but not necessarily large), thus requires more explanation. 
2) Avoiding strict saddles. R1 clarified that he was not claiming "[17, 18] or Pemantle alone already covered this paper’s result", but asking whether a simple combination of [17, 18] and Pemantle [27] would give their result (that’s why R1 used “morally” in the review). Of course this is a valid question. The rebuttal argued “in the stochastic case, the presence of a non-trivial center manifold requires fundamentally different techniques from differential geometry, as we explain in detail in Appendices D.2 and D.3”, which I think provide a reasonable answer to this question. All reviewers still appreciate the technical contribution of this part, and it is also the main reason of acceptance. 3) Covnergence rate to local min. R3 pointed out three limitations: a) strict local-min with PD Hessian; b) a local rate result; c) hidden dependence on m. The first restriction is especially important. The rebuttal highlighted the improvement of 1/sqrt{n} in [14] to 1/n rate in this paper. During discussion, R1 pointed out that the improvement over [14] is an unfair comparison. More specifically, “O(1/\sqrt{n}) results in [14] is for finding Second-Order-Stationary-Point (SOSP),… this paper assumes locally strongly convex local-min. Their O(1/n) rate should be a direct consequence of this local strongly convexity assumption. Theorem 5 in [14] also gives the result for the local strongly convex setting, since [14] studies the deterministic gradient case, it can even achieves e^{-n} rate.“ The authors shall highlight "strict local-min" or "locally strongly convex local-min" whenever mentioning this convergence result. R2 questioned the advantage over Bonnabel (2013). I think results on compact Riemann manifold do not directly imply results in Euclidean space and might require substantial different techniques, thus the existence of Bonnabel (2013) is not a major issue. But as R1 suggested, please give more credit to Bonnabel and Permantle since they resolved partially the same problem (morally). Reviewers agree that this paper proved some new results on a problem of interest to the community, and the merits outweigh the pitfalls, so I recommend accept. Nevertheless, please modify the paper accordingly to soften the claims on the contributions on boundedness assumption and convergence rate result; e.g., in the introduction, the authors shall highlight they only provide a partial answer to Question 3 and 1, since these two parts are significantly weaker than Part 2.